# Corrosion Resistance of CeO_2_-GO/Epoxy Nanocomposite Coating in Simulated Seawater and Concrete Pore Solutions

**DOI:** 10.3390/polym15122602

**Published:** 2023-06-07

**Authors:** Xiaoyan Liu, Zitao Wu, Yaoyao Lyu, Tianyu Li, Heng Yang, Yanqi Liu, Ruidan Liu, Xian Xie, Kai Lyu, Surendra P. Shah

**Affiliations:** 1College of Mechanics and Materials, Hohai University, Nanjing 210098, China; 2Institute of Corrosion Protection, Hohai University, Nanjing 210098, China; 3College of Water Conservancy and Hydropower Engineering, Hohai University, Nanjing 210098, China; 4Materials & Structural Engineering Department, Nanjing Hydraulic Research Institute, Nanjing 210024, China; 5College of Civil and Transportation Engineering, Hohai University, Nanjing 210098, China; 6Department of Civil Engineering, The University of Texas at Arlington, 701 S. Nedderman Drive, Arlington, TX 76019, USA

**Keywords:** graphene oxide, cerium dioxide, nanocomposite coating, simulated concrete pore liquid, erosion resistance

## Abstract

Reinforced concrete structures in the marine environment face serious corrosion risks. Coating protection and adding corrosion inhibitors are the most economical and effective methods. In this study, a nano-composite anti-corrosion filler with a mass ratio of CeO_2_:GO = 4:1 was prepared by hydrothermally growing cerium oxide on the surface of graphene oxide. The filler was mixed with pure epoxy resin at a mass fraction of 0.5% to prepare a nano-composite epoxy coating. The basic properties of the prepared coating were evaluated from the aspects of surface hardness, adhesion grade, and anti-corrosion performance on Q235 low carbon steel subjected to simulated seawater and simulated concrete pore solutions. Results showed that after 90 days of service, the corrosion current density of the nanocomposite coating mixed with corrosion inhibitor was the lowest (I_corr_ = 1.001 × 10^−9^ A/cm^2^), and the protection efficiency was up to 99.92%. This study provides a theoretical foundation for solving the corrosion problem of Q235 low carbon steel in the marine environment.

## 1. Introduction

For industrial development and market demand, steel-reinforced concrete is widely applied as an engineering material in the construction of a range of offshore infrastructure such as ports and coastal, confederation bridges, and transport pipelines, which are increasingly being put into production and operation [1]. However, seawater corrosion can lead to problems such as surface cracking, declining load carrying capacity, and lower life expectancy of concrete-reinforcing bars [2,3,4,5], and steel corrosion poses a serious risk to economic safety [6]. For this reason, the study on corrosion protection of concrete-reinforcing bars under marine environment can minimize economic losses and promote the development of marine resources [7,8,9]. Reinforcing bar corrosion in the marine environment is one of the major challenges, and it is therefore of great significance to study its protection measures.

Traditional corrosion protection methods such as concrete surface treatment, coating protection, electrochemical cathodic protection of metals, and the addition of corrosion inhibitors can inhibit the corrosion of concrete-reinforcing bars [10,11,12]. Of these methods mentioned above, coating protection is favored due to its high corrosion protection efficiency, simplicity of process, and low cost [13,14]. Among all kinds of coating anti-corrosion technologies, epoxy coatings are of great interest due to their excellent working properties and simple application process [15,16]. Previous studies have shown, however, that epoxy coatings contain hydrophilic groups that may form water seepage channels in harsh environments [17,18] and that epoxy coatings are brittle and prone to cracking when exposed to corrosive environments for long periods of time [19,20]. In an effort to overcome the poor durability of epoxy coatings, a considerable number of scholars in the field of coating corrosion protection have extensively examined complex coatings reinforced with polymers in recent years and have made great achievements in the field of epoxy coating corrosion protection. However, the complexity and high cost of the process has limited the use of polymer fillers in corrosion protection on a large scale. To address this issue, many researchers have dedicated themselves to developing a large number of alternative materials to facilitate the application of epoxy coatings in the corrosion field. Among these, graphene, with its high tensile strength (130 GPa) [21], is an effective reinforcement material in polymer matrices and has attracted extensive research interest among scholars in the field of corrosion due to its natural stratified structure [22]. For example, Tian et al. modified GO with quaternary ammonium compounds to enhance the anti-corrosion performance of epoxy coatings. [23]. Huang et al. designed a new graphene composite coating with great potential for corrosion and wear resistance of titanium alloys using cathodic electrophoretic deposition [24]. Ramezanzadeh et al. [25] grafted 3-aminopropyltriethoxysilane and graphene oxide into a silicon-containing epoxy resin to enhance the corrosion protection properties of the epoxy coating. SONG R [26] et al. prepared nanocomposites by pulse plating, which showed excellent resistance to seawater corrosion. Laminar graphene is embedded inside organic coatings and is widely used in solvent-based anticorrosion coatings [27], acting as a barrier to corrosive medium [28,29]. In available research, however, the problem of easy agglomeration of graphene oxide has not been addressed, and many studies have focused on the physical barrier effect of graphene oxide in polymer coatings, with less research on the repairing effect of graphene oxide in coating corrosion protection.

As a filler, graphene oxide can improve the mechanical strength and corrosion resistance of polymer coatings. However, there is little research in the existing literature on improving the dispersibility of graphene oxide in polymer coatings. Cerium oxide (CeO_2_) is a rare-earth metal oxide with strong oxidizing properties [30] and its hydrolysis product is a white precipitate. To date, several studies have shown that cerium oxide is used as an additive in the glass industry for flat glass grinding [31,32]. In the field of optoelectronics, cerium oxide shows anti-UV effects [33,34]. However, in marine environments, the corrosion protection of cerium oxide as an anti-corrosion material against steel bars in concrete pore solutions is still rarely studied. Due to the advantages of graphene oxide and cerium oxide, an anticorrosive filler with a mass ratio of CeO_2_:GO = 4:1 was prepared by hydrothermal synthesis, and nanocomposite anticorrosion coatings were prepared by nano-fillers in epoxy resin in this paper. Corrosion protection performance of nanocomposite anti-corrosion coatings on concrete-reinforcing bars in the marine environment was then investigated by comparison with corrosion inhibitors.

According to our previous research, cerium oxide-modified graphene oxide nanocomposites (CeO_2_:GO (4:1)) prepared by hydrothermal synthesis [35] had excellent corrosion resistance in simulated seawater, simulated saline-alkali, and simulated acid rain environments [36]. Marine construction represented by reinforced concrete structures are subject to serious corrosion risks, and the aim of this paper is to further investigate the corrosion performance and mechanical properties of CeO_2_-GO/EP nanocomposite coatings in simulated seawater and simulated concrete pore liquid composite solutions. The basic properties of the prepared coating were evaluated from the aspects of surface hardness and adhesion grade, as well as its anti-corrosion performance that was subjected to simulated seawater and simulated concrete pore liquid composite solutions by electrochemical tests. Compared to previous research, the innovation of this study lies in the application of CeO_2_-GO nanocomposites to epoxy coatings and the exploration of their corrosion resistance in simulated seawater and concrete pore solutions, as well as the application prospects for corrosion protection of concrete structures. The significance of this work is to expand the application of this composite material and provide a new solution for the corrosion protection of concrete buildings. Compared with the pure epoxy resin, The results shows that the hardness and adhesion of CeO_2_:GO (4:1) nanocomposite coating increased by nearly 50% compared with pure epoxy coating after 30 days of service. After 60 days of service, the corrosion degree of steel bar decreased significantly, and the corrosion area decreased from 41.4% to 22.8%. The electrochemical impedance value increased by three orders of magnitude. After 90 days of service, the corrosion current density of the nanocomposite coating mixed with corrosion inhibitor was the lowest (I_corr_ = 1.001 × 10^−9^ A/cm^2^), and the protection efficiency was up to 99.92%.

## 2. Experimental

### 2.1. Raw Materials

Graphene oxide was supplied by Changzhou Sixth Element Materials Technology Co., Changzhou, China. Cerium nitrate hexahydrate was purchased from Shanghai Aladdin Biochemical Technology Co., Shanghai, China. Epoxy resin (WSR6101 E-44) and epoxy AB adhesive were supplied by Nantong Xingchen Synthetic Materials Co, Nantong, China. The epoxy resin (WSR6101 E-44) is a bisphenol thermosetting epoxy resin with a viscosity of 15,000 mPa·s at 25 °C. The selected Lanxing Lan-826 multifunctional pickling corrosion inhibitor was purchased from Henan Xinyang Chemical Co, Xinyang, Henan Province, China. CaCl_2_, NaF, KCl, NaHCO_3_, KBr, H_3_BO_3_, NaCl, and Na_2_SO_4_ were supplied by Chengdu Kolon Chemical Co., Chengdu, China. Table 1 describes the chemical composition of Q235 steel sheet. The basic properties of Lan-826 corrosion inhibitor are shown in Table 2.

### 2.2. Preparation of CeO_2_-GO (4:1)/EP Nanocomposite Coating

CeO_2_-GO nanocomposites were prepared by hydrothermal synthesis method and mixed with epoxy resin to prepare CeO_2_-GO/EP composite coatings. In 500 mL of deionized water, 10 g of sodium bicarbonate and 186.6 mg of ethylenediaminetetraacetic acid disodium salt dihydrate (EDTA.2Na) were dissolved. The solution was stirred well and placed in a dialysis bag. The solution in the dialysis bag was boiled for 10 min. After the dialysis bag was cleaned, the graphene oxide liquid was added, and the mixture was boiled again in a beaker for 10 min. Following that, the liquid was dissolved in deionized water and dispersed by ultrasonic treatment for 30 min. The dispersed solution was transferred into the dialysis bag and stirred in a beaker for three days. Finally, the liquid was extracted from the dialysis bag and a graphene oxide dispersion was obtained using an ultrasonic cell disrupter. 0.807 g of cerium nitrate hexahydrate was weighed and dissolved in a beaker containing 4 mL of ammonia solution. The solution was stirred with a glass rod and then transferred into a polytetrafluoroethylene-lined container, which was placed inside a high-pressure reactor. The reactor was placed in an oven set at 160 °C for 24 h. After the reaction was complete, the product was filtered, dried, and ground to obtain CeO_2_-GO nanoparticles. In the experiments, Q235 steel sheets(150 mm × 70 mm × 1 mm) were used to assess the hardness and adhesion of the coating, and Q235 mild steel with dimensions of 5 mm in height and 10 mm in diameter was selected to evaluate the anticorrosive properties of the coating. The prepared CeO_2_-GO composite material was mixed with epoxy resin in a mass ratio of 0.5% and coated on the surface of Q235 steel through a wire rod coating device. After the coating solidified, it was rubbed with sandpaper to ensure that the coating thickness was controlled at 100 μm by a coating thickness gauge. The samples were immersed in simulated seawater and concrete pore liquid composite solutions for 90 days to test the coating’s corrosion resistance.

For comparison, as shown in Table 3, four epoxy coating samples, G1, G2, G3, and G4, were prepared, with G1 being the pure epoxy coating, G2 being the epoxy coating with corrosion inhibitor, G3 being the CeO_2_-GO/EP coating, and G4 being the CeO_2_-GO/EP coating with corrosion inhibitor. The anti-corrosion performance of the four coatings subjected to simulated seawater and simulated concrete pore liquid composite solutions were evaluated.

### 2.3. Preparation of Corrosive Solutions

Different concentrations of simulated seawater and simulated concrete pore liquid composite solutions (20%, 40%, 60%) were prepared according to ASTM D1141-1998 (2013). As shown in Figure 1, 24.53 g of NaCl and 4.09 g of Na_2_SO_4_ were dissolved in 800 mL of the aqueous solution. We slowly added 20 mL of stock solution No. 1 with vigorous stirring, then added 10 mL of stock solution No. 2. The solution pH was adjusted to 12.5 with Ca(OH)_2_ after dilution to 1 L. Table 4 shows the chemical composition of the simulated seawater and simulated concrete pore solution.

### 2.4. Test Method

#### 2.4.1. Corrosion Micromorphology Testing

To visualize the microstructure of CeO_2_-GO nanocomposite coatings in the seawater and concrete pore liquid environments with long-term service, a scanning electron microscope (FESEM, Gemini, M/s.Zeiss, Karlsruhe, Germany) was used with a working distance of 7 mm and an acceleration voltage of 1 kV at room temperature. With pure epoxy coating as a contrast, G1 and G3 samples were immersed in 60% hybrid solution for 60 days to characterize the micromorphology of the corroded coating substrate and to evaluate the anti-corrosion effect of CeO_2_-GO (4:1) nanocomposites. When calculating the erosion area of the sample, we selected the observation area and determined its size and shape. The size of the observed area can be measured using the eyepiece scale and objective scale of a microscope. Based on the measured shape and size, the area of the corroded part of the sample can be calculated. 

#### 2.4.2. Hardness Test

The hardness of the coatings was tested according to standard ISO 15184:1998 [37]. The hardness of a coating was determined using the pencil scratch hardness test. The hardness of the hardest pencil that does not leave a scratch exceeding 3 mm on the coating was used to represent the hardness of the coating. The higher the hardness of the coating, the better the wear resistance of the coating. The coating samples were evenly coated with (20 ± 3) μm thickness on a Q235 steel surface of 150 mm × 70 mm × 1 mm, respectively. After the samples were cured at room temperature for 24 h, the hardness of the coatings were measured at a temperature of (23 ± 2) °C and relative humidity of (50 ± 5)%. All samples were immersed in different concentrations of the solutions simultaneously for 15 days to obtain the hardness of the coatings after service in the simulated seawater and simulated concrete pore liquid composite solutions.

The adhesion values of the coatings were determined according to standard ISO 2409:2013 [38]. The sample preparation is the same as the hardness test. The coating samples were evenly coated with (20 ± 3) μm thickness on a carbon steel surface of 150 mm × 70 mm × 1 mm, respectively. After each sample was immersed in the solution for 30 days, the coating adhesion tester was used to conduct the test under the conditions of temperature (23 ± 2) °C and relative humidity (50 ± 5)%. First, the cured coating template was fixed in a position perpendicular to a grid device, and a cross-shaped scratch was made on it. Then, a transparent tape was uniformly covered over the scratch. Finally, the tape was removed, and the adhesion strength was determined by observing whether the scratch comes off or not according to certain criteria. As shown in Table 5, the adhesion grade was divided into six grades according to the area of the coating surface peeling off. The smaller the grade, the smaller the area of the coating peeling off, and the better the adhesion.

#### 2.4.3. Electrochemical Test

Electrochemical tests were performed using a CHI-760E electrochemical workstation with a three-electrode system. The platinum electrode and saturated glycury electrode were used as counter electrode and reference electrode, respectively. G1, G2, G3, and G4 were conducted as working electrodes after the open circuit potential values were stabilized. The exposed area of the working electrode is 1 square centimeter. In electrochemical experiments, the thickness of all coatings is controlled at 100 μm. In EIS testing, the initial potential (Init E (V)) is the stable value of the open circuit potential, the high frequency (Hz) is 100,000, the low frequency (Hz) is 0.01, the amplitude (V) is 0.005, and the quiet time (sec) is 2.

After the samples were immersed in the composite solution for 90 days, the tafel polarization curves were recorded at a scanning speed of 0.5 mV/s in the range of −200 mV to +1200 mV to investigate the trends of kinetic parameters of the coatings after long-term service in simulated seawater and simulated concrete pore liquid composite solutions with different concentrations. Table 6 describes the time when all electrochemical tests were done. The EIS and Tafel tests were completed simultaneously within 90 days. In the EIS testing, as there was no significant difference between the 90 day test results and the 60 day test results, in order to avoid data redundancy, Section 3.4.2 only analyzed the EIS test results within 60 days.

## 3. Results and Discussion

### 3.1. SEM-EDS Analysis of CeO_2_-GO Nanocomposites

Figure 2 and Figure 3 show the SEM corrosion morphologies of G1 and G3 after 60 days of service in a 60% corrosive solution. According to Figure 2a, the surface of the epoxy coating is cracked. The pure EP coating has a severe degree of corrosion, and there are deep corrosion pits due to large-scale peeling of the coating. From Figure 2b–d, it can be seen that G1 shows a large area of rough and concave corrosion pits, and a large number of corrosion products exist on the surface. The corrosion area of the substrate was observed through an optical microscope, with the black shaded area being the rusted portion and the white area being the uncorroded portion. Figure 2e and Figure 3e show the macroscopic morphology of G1 and G3 after 60 days of corrosion, respectively. Calculations revealed that the erosion portion of G3 was 22.8%, which was only half that of the G1. As shown in Figure 3b–d, the uniform corrosion of G3 coating occurred because the graphene oxide modified by cerium oxide was uniformly distributed in the epoxy resin in the form of lamellae, filling the microporous defects of the pure epoxy resin. The scanning electron microscopy and energy spectrum analysis results of CeO_2_-GO (4:1) (Figure 4) revealed that CeO_2_ was successfully grafted onto the surface of GO, indicating the successful preparation of CeO_2_-GO (4:1) nanocomposite material.

### 3.2. Hardness

Figure 5 shows the hardness test results for all coatings after 30 days of immersion in different concentrations of the simulated mixture. The hardness value of the coating doped with CeO_2_-GO nanocomposite material was nearly twice that of the pure EP coating, which indicates that the graphene oxide with excellent mechanical properties filled the microporous defects in the pure epoxy resin and improved the hardness of the pure epoxy resin [39]. As shown in Figure 6, as the concentration increases, the hardness values of G1 and G2 decreased significantly, while that of G3 and G4 decreased slightly. The hardness values of G1, G2, G3, and G4 coatings were in descending order of magnitude. The reason is that there were many bubbles and pores on the surface and inside of the pure EP coating, and more corrosion solutions immersed in the coating, causing the surface of the coating to expand and crack. The hardness value of the same sample decreased with the increase of solution concentration, demonstrating that the higher the concentration of the solution, the more quickly the corrosive medium penetrated the coating, and the faster the hardness of the coating decreased. As can be seen in Figure 6, the addition of corrosion inhibitors can slightly delay the rate of decline in the hardness value of the coating, but its effect was not apparent.

### 3.3. Adhesive Force

The results of the adhesion test after 30 days of serviced in the simulated seawater/concrete pore liquid composite solution environment are presented in Figure 6. G3 and G4 had the lowest adhesion grade and the smallest spalling area, indicating that CeO_2_-GO (4:1) nanocomposites are fully bonded with epoxy resin and CeO_2_-GO/EP has good stability. Compared with CeO_2_-GO/EP coating, pure EP coating has poor crack resistance due to the loose and porous surface, which makes the corrosion ions penetrate continuously. After 30 days of service, the adhesion grades of G1 and G2 decreased significantly, while the adhesion of G3 and G4 coatings was still larger. As shown in Figure 7, after 30 days of service, the adhesion values of G1 and G2 coatings decreased significantly, while that of G3 and G4 decreased slightly. It can be seen that CeO_2_-GO (4:1) nanocomposites effectively improved the adhesion of the coating, which was consistent with the hardness test results.

### 3.4. Electrochemical Testing

#### 3.4.1. Open Circuit Potential Test

Figure 7 shows the time-dependent curves of the open-circuit potential of coatings (G1, G2, G3, G4) during the 60-day service of simulated seawater and simulated concrete pore–liquid mixed solution with different concentrations. The results show that the open circuit potentials of G3 and G4 in 20% and 40% hybrid solution were less than −200 mV when soaking for 15 days, indicating that CeO_2_-GO/EP effectively blocked the erosion of corrosive ions. The open circuit potential values of G1 and G2 are close to −400 mV, meaning that the corrosive medium has penetrated inside the coating, and the substrate has started to corrode. With the extension of coating service time, the open circuit potential of G1 and G2 decreases sharply, and the decrease of G3 and G4 is relatively tiny. The open circuit potential values of coatings show the same trend (G4 > G3 > G2 > G1) at each stage of service in the same corrosion environment, which may be due to the physical barrier of CeO_2_-GO nanocomposite and the release of cerium ions. After 45 days of service, the open-circuit potential of the coatings in different solutions no longer changed significantly, and the corrosion entered the middle and late stages when the physical shielding ability of the coatings began to fail.

#### 3.4.2. Electrochemical Impedance Spectroscopy

Figure 8 and Figure 9 show the Nyquist diagrams of each coating during 60 days of service in simulated seawater and concrete pore liquid composite solutions with different concentrations. Figure 8 describes the Nyquist diagrams of coatings immersed in composite solutions with different concentrations (20%, 40%, 60%) for 30 days. Each coating showed a large capacitive arc in the low-frequency zone, indicating that the coating is in the early immersion stage. At this time, the epoxy resin was well isolated from the corrosive ions in the composite solution due to its strong adhesion ability and good electrical insulation. It is worth noting that, as shown in Figure 8c, in the 60% concentration composite solution, G1 presented two capacitive arcs, which indicates that the pure EP coating has entered the middle stage of corrosion. The corrosion medium has gradually penetrated the substrate surface through the pores and defects of the coating. As shown in Figure 8f, after the coating had served in the 60% solution for 30 days, the second half capacitive arc appeared in G2 and G3, and G1 completely formed two capacitive arcs. Although the Nyquist diameter of the G4 reduced, it still appeared as a capacitive arc. It can be found that the G4 coating was in the initial stage of corrosion, and other coatings were in the middle stage of corrosion.

Figure 9 shows the Nyquist diagrams of each coating immersed in the hybrid solution for 60 days, and the capacitive arc radius of the curve decreased regularly. Figure 9a shows the Nyquist diagrams of the coatings after 45 days of service in a 20% composite solution. The first capacitive arc of G1 and G2 further decreased, and G3 coating presented a complete capacitive arc and a second capacitive arc, which indicates that G3 began to form defects. At this time, G4 still showed good corrosion resistance due to the inhibition of corrosion inhibitors. In Figure 9b, the capacitive arc radius of G1, G2, G3, and G4 presented two arcs from small to large, which indicates that all coatings began to enter the middle stage of corrosion after 45 days of service in a 40% hybrid solution. Although the capacitive arc radius of the coatings still decreased in the later period of immersion, the rate decreased. It can be seen from Figure 9e that, after 60 days of service in a 40% hybrid solution environment, G1 appeared Warburg tail in the low-frequency zone, beginning to fail. As the corrosion medium continued to diffuse to the interior of the substrate, G2, G3, and G4 showed a capacitive arc. The G2, G3, and G4 all show a capacitive arc due to the passive film and corrosion products generated in the concrete pore liquid environment, delaying the metal corrosion rate.

The Bode diagrams of coatings (G1, G2, G3, G4), immersed in simulated seawater and simulated concrete pore liquid-composite solution for 60 days, are shown in Figure 10 and Figure 11. It can be seen from Figure 12 that the impedance value of each coating in different solutions was considerable when soaked for 15 days, and the impedance modulus values in the low-frequency region of all coatings show the same rule: G4 > G3 > G2 > G1. After soaking in 20% composite solution for 15 days, the low-frequency impedance modulus of G4 coating exceeds 10^8^ Ω cm^2^, and the low-frequency impedance modulus of each coating decreases with the increase of corrosion solution concentration. Figure 10a,c show that the low-frequency impedance modulus of G1 and G2 decreased significantly with the increase of seawater concentration. When the solution concentration increases from 20% to 60%, the low-frequency impedance modes of G1 and G2 decrease from 10^7^ Ω cm^2^ to 10^6^ Ω cm^2^, while the low-frequency impedance modes of G3 and G4 coatings decrease slightly. This phenomenon indicated that CeO_2_-GO/EP inhibited the corrosion rate of metals. As the immersion time increased, the corrosion resistance of the coating gradually degraded. As shown in Figure 10e and Figure 11b, the low-frequency impedance modulus of G2 and G1 decreased from 10^6^ Ω cm^2^ to 10^5^ Ω cm^2^ after immersion in 40% hybrid solution for 30 d and 45 d. It can be seen from Figure 11 that after 60 days of service, the impedance values of all coatings continue to decrease, but the decrease is slight. The reason is that the coatings almost fail in the later stage of corrosion, which is consistent with the analysis results of the Nyquist diagram.

#### 3.4.3. Tafel Curve Analysis

The Tafel curve represents the strongly polarized part of the polarization curve. By studying the relationship between corrosion current density and corrosion voltage, the polarization resistance and protection efficiency of anti-corrosion coatings were analyzed [40]. The expression is as follows.
(1)$$R_{p}=\frac{\beta_{a}\beta_{c}}{2.303\left(\beta_{a}{+\beta }_{c}\right)I_{corr}}$$
(2)$$\eta \;\%=\left(1-\frac{I_{corr}}{I_{corr0}}\right)\times 100\%\;$$

$\beta_{a}$ represents the slope of the Tafel curve of the electrode anode. $\beta_{c}$ represents the slope of the Tafel curve of the electrode cathode. $I_{corr}\;$represents the current corrosion density of the anti-corrosion coating. $I_{corr0}$ represents the current corrosion density of a pure epoxy resin coating. $R_{p}$ is the coating polarization resistance of the prepared electrode. η represents the coating protection efficiency.

Figure 12 shows the Tafel curves of the coating soaked in simulated seawater and simulated concrete pore–liquid/composite solution for 90 days. Table 7 shows the corrosion kinetic parameters of the coating soaked in a 20% concentration hybrid solution for 90 days. The corrosion current density of G2 is 2.143 × 10^−7^ A/cm^2^, which is smaller than that of EP coating. The corrosion current density of G4 is only 1.001 × 10^−9^ A/cm^2^, slightly smaller than that of G3, and its protection rate reaches 99.92%. In comparison with G1, the corrosion current density of G4 was reduced by three orders of magnitude. The polarization resistance of G4 was the largest, which was 9.473 × 10^8^ ohm, which was slightly larger than that of G3. The corrosion inhibitor could boost the erosion resistance of the metal, but the effect is not apparent. After 90 d in the 20% concentration compound solution, the relationship between the magnitude of corrosion current density is G4 < G3 < G2 < G1, while the relationship between the magnitude of corrosion potential of the coating shows the opposite rule: G4 > G3 > G2 > G1, which shows that CeO_2_-GO nanomaterials can effectively improve the corrosion protection performance of the coating. According to Table 8 and Table 9, the corrosion current density of the coating increased with the ascension of solution concentration. Meanwhile, the corrosion potential and protection efficiency decreased. However, after immersion in 60% hybrid solution for 90 days, the protection efficiency of G4 is still as high as 94.76%. Compared with G1, the protection efficiency of G4 increased by 33.48%. It is proved again that CeO_2_-GO/EP can effectively improve the corrosion resistance of metal in the seawater and concrete pore liquid composite solutions environment.

The protection efficiency of the coatings after 90 d of service in pure simulated seawater and simulated seawater and simulated concrete pore liquid composite solutions, respectively, is shown in Figure 13. In the same concentration of the corrosive solution, the protection efficiency of the coating in the composite solution environment is more effective than that in the pure simulated seawater environment. It is worth noting that the self-corrosion current density of pure EP in the composite solution environment is much lower than that in the pure simulated seawater environment, indicating that the coating has better corrosion protection under concrete pore liquid conditions.

### 3.5. Discussion

The graphene oxide nanocomposites modified by CeO_2_ can be fully dispersed in the epoxy resin, filling the microporous defects of pure epoxy resin and improving the mechanical properties of the coating. Compared with the pure epoxy resin, the results show that the hardness and adhesion of CeO_2_:GO (4:1) nanocomposite coating increased by nearly 50% compared with pure epoxy coating after 30 days of service. After 60 days of service, the corrosion degree of steel bar decreased significantly, and the corrosion area decreased from 41.4% to 22.8%. The electrochemical impedance value increased by three orders of magnitude. After 90 days of service, the corrosion current density of the nanocomposite coating mixed with corrosion inhibitor is the lowest (I_corr_ = 1.001 × 10^−9^ A/cm^2^), and the protection efficiency is up to 99.92%. On the one hand, the physical barrier effect of CeO_2_-GO nanocomposite dramatically improves the corrosion resistance of the coating. On the other hand, hydrates formed by hydrolysis of cerium ions also inhibit the corrosion reaction. The schematic diagram of the coating corrosion process is shown in Figure 14a. The loose and porous surface of the pure EP coating allows for the continuous penetration of corrosion ions. As a result, the internal water absorption of the pure epoxy coating becomes more prominent, and thus the corrosion rate is faster. The CeO_2_-GO/epoxy nanocomposite coating possesses self-repair capability. This mechanism is based on the interaction between two key materials: cerium oxide and graphene oxide. When the surface of the coating sustains minor damage, the nanoparticles of CeO_2_ and GO can diffuse and fill the defects and cracks, reacting with the surrounding substrate to form a new protective layer [41]. In this way, CeO_2_-GO can partially repair itself when subjected to corrosion through the interaction between cerium ions and graphene oxide, thereby maintaining its anti-corrosion properties. This “corrosion self-repair coordination mechanism” provides more reliable and long-lasting protection for anti-corrosion materials, enabling them to be used in more severe environments [42].

Figure 14b is a schematic diagram of the corrosion process of CeO_2_-GO/EP. Cerium oxide is adsorbed on the lamellar graphene oxide surface by hydrothermal synthesis, making graphene oxide nanosheets uniformly dispersed in the epoxy coating and filling microporous defects. As the coating enters the middle and later stages of corrosion, as shown in Figure 15, the corrosion solution penetrates the coating and contacts the substrate. Meanwhile, Ce^4+^ can be rapidly released into the corrosion solution due to the cation exchange mechanism [43]. Cerium oxide hydrolyzes in the corroded cathode area, producing a water-soluble precipitate layer that covers the substrate surface and inhibits corrosion in the cathode area. Additionally, as the concrete pore liquid is alkaline, the passive film is formed on the surface of the substrate, which cooperates with the corrosion products to jointly resist the further corrosion of corrosion ions.

## 4. Conclusions

In this paper, CeO_2_:GO (4:1) nanocomposites were produced by hydrothermal synthesis, and CeO_2_-GO/EP coatings were formulated by being mixed with epoxy resin. This paper investigates the corrosion behavior of steel reinforcement in offshore structures under different concentrations of marine environments. The study involves introducing simulated concrete pore liquid into simulated seawater to simulate the actual corrosive conditions. The corrosion resistance and mechanical properties of CeO_2_-GO/EP in simulated seawater/concrete pore liquid composite solutions with different concentrations were investigated. The corrosion areas of the coatings after 60 d of service were calculated by optical microscopy, and the corrosion area of CeO_2_:GO (4:1)/EP was reduced by half compared to that of pure epoxy (41.4%), with a value of 22.8%. The evolution of the corrosion state of the coatings was investigated by electrochemical tests during 90 d, indicating that the low-frequency impedance value of CeO_2_:GO (4:1)/EP doped with corrosion inhibitor was the largest (|Z|0.01 = 10^8^ ohm cm^2^). By calculating Tafe polarization data, the corrosion current density of CeO_2_:GO (4:1)/EP after 90 days of service was I_corr_ = 1.435 × 10^−8^ A/cm^2^. After calculation and analysis, the protection efficiency of CeO_2_:GO (4:1)/EP was 98.91%, significantly improving the corrosion resistance of the coating.

## Figures and Tables

**Figure 1 polymers-15-02602-f001:**
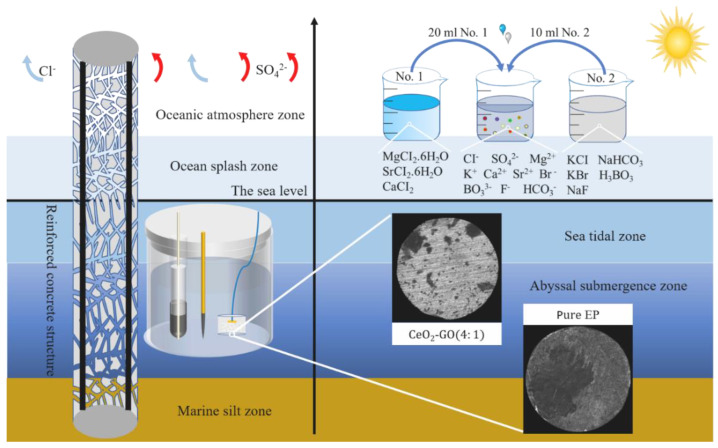
Schematic diagram of preparation of corrosive solution: preparation of corrosive solution and optical microscope morphology after corrosion.

**Figure 2 polymers-15-02602-f002:**
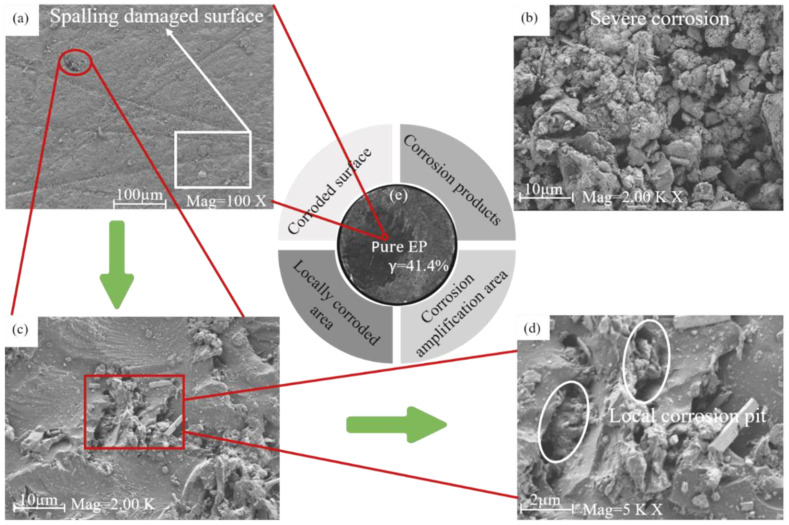
SEM images of EP in 60% corrosive solution for 60 days: (**a**) corroded surface; (**b**) corrosion products; (**c**) locally corroded; (**d**) corrosion amplification area; (**e**) optical microscopic morphology after 60 days of corrosion.

**Figure 3 polymers-15-02602-f003:**
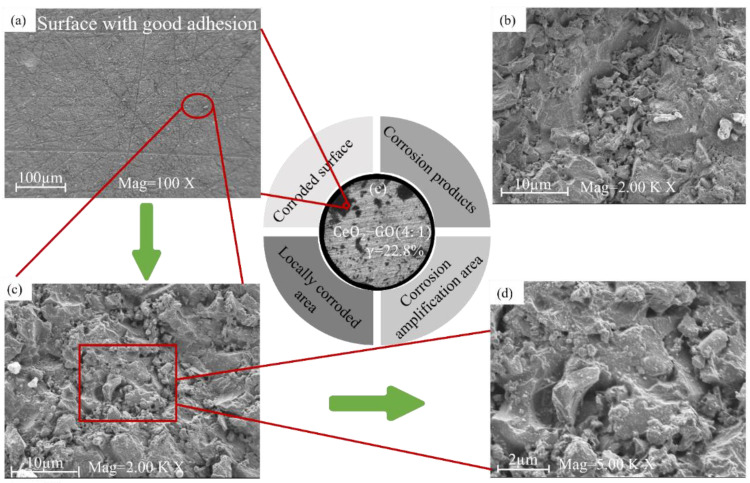
SEM images of CeO_2_-GO/EP in 60% corrosive solution for 60 days: (**a**) corroded surface; (**b**) corrosion products; (**c**) locally corroded; (**d**) corrosion amplification area; (**e**) optical microscopic morphology after 60 days of corrosion.

**Figure 4 polymers-15-02602-f004:**
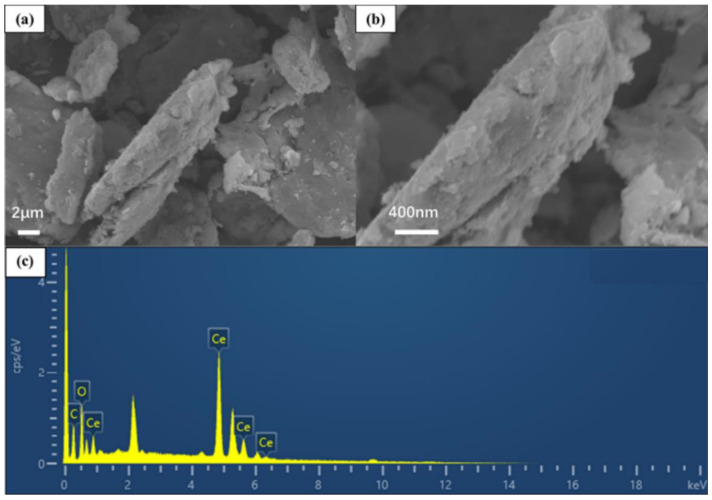
Scanning electron microscopy and energy spectrum analysis of CeO_2_-GO(4:1): (**a**,**b**) show the microstructure of CeO_2_-GO nanocomposites; (**c**) shows the energy spectrum of CeO_2_-GO nanocomposites.

**Figure 5 polymers-15-02602-f005:**
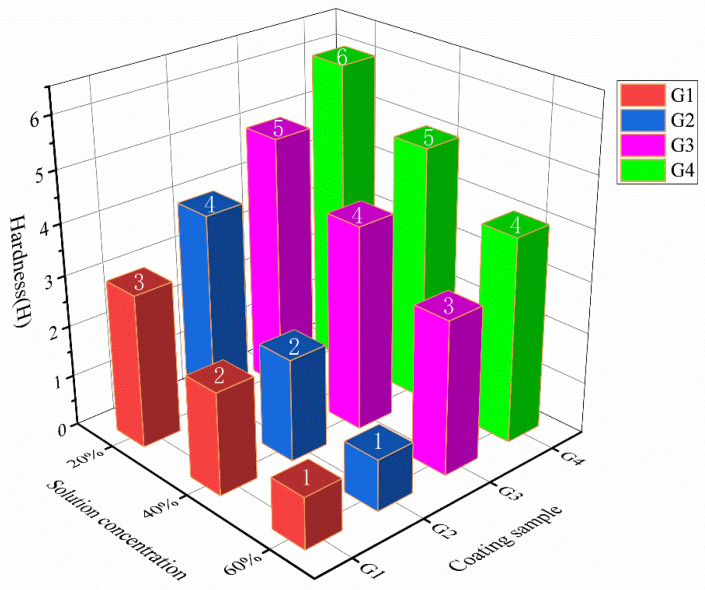
Hardness changes of coatings after 30 d of service in different concentrations of composite solution: The numbers in the figure are the hardness values of the coating.

**Figure 6 polymers-15-02602-f006:**
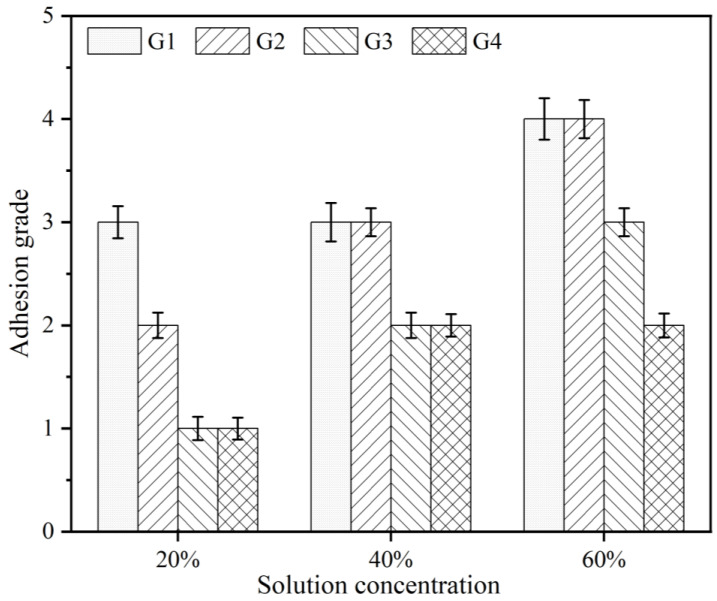
Adhesion grade of coatings after 30 days of service in different concentrations of composite solution.

**Figure 7 polymers-15-02602-f007:**
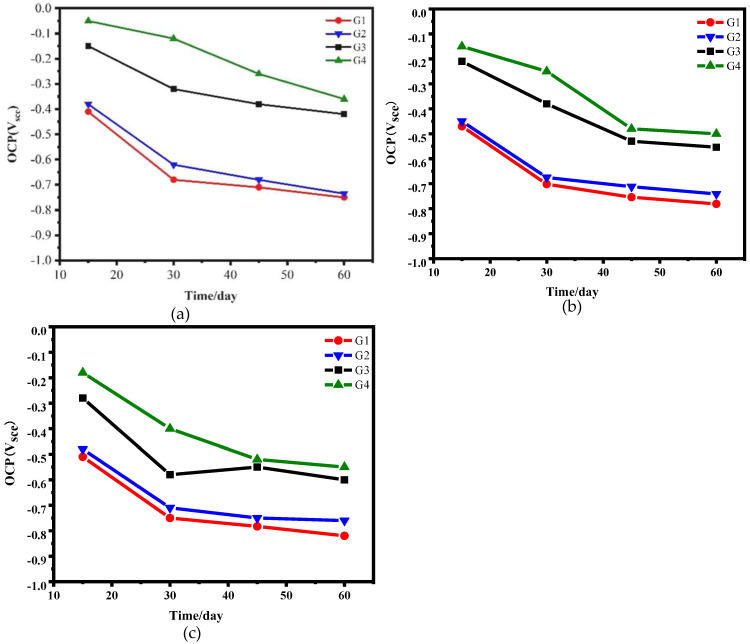
The OCP of each coating during 60 days of service in different concentration solutions: (**a**) 20%; (**b**) 40%; (**c**) 60%.

**Figure 8 polymers-15-02602-f008:**
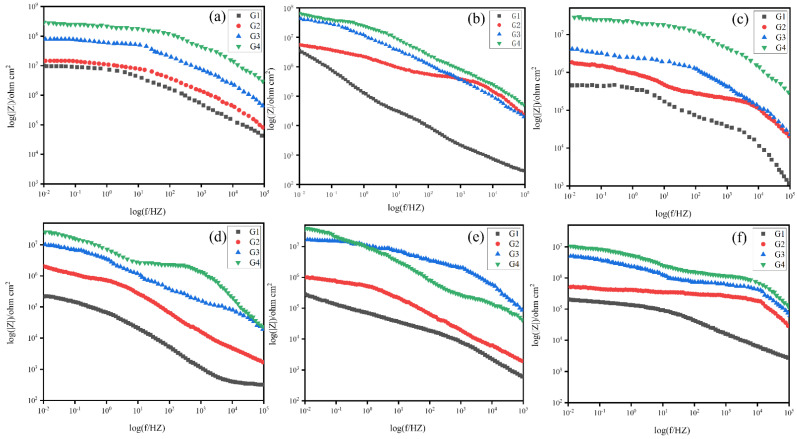
Nyquist plots of all coatings served for 30 days in corrosive solutions of different concentrations: (**a**–**c**): After 15 days of service; (**d**–**f**): After 30 days of service.

**Figure 9 polymers-15-02602-f009:**
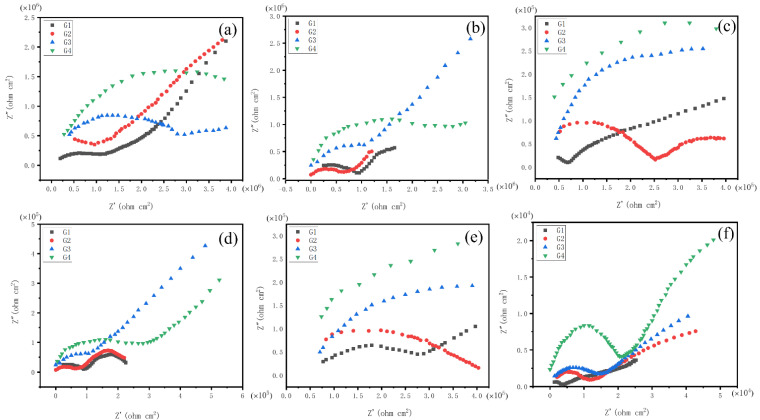
Nyquist plots of all coatings served for 60 days in corrosive solutions of different concentrations: (**a**–**c**): After 45 days of service; (**d**–**f**): After 60 days of service.

**Figure 10 polymers-15-02602-f010:**
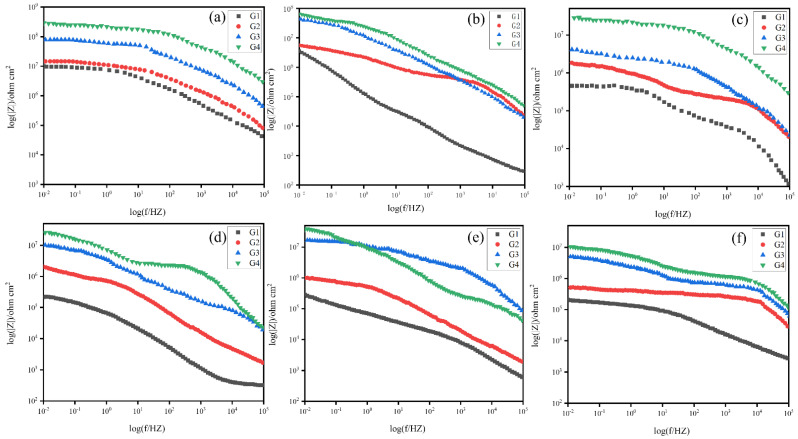
Bode diagram of all coatings served for 30 days in corrosive solutions of different concentrations: (**a**–**c**): After 15 days of service; (**d**–**f**): After 30 days of service.

**Figure 11 polymers-15-02602-f011:**
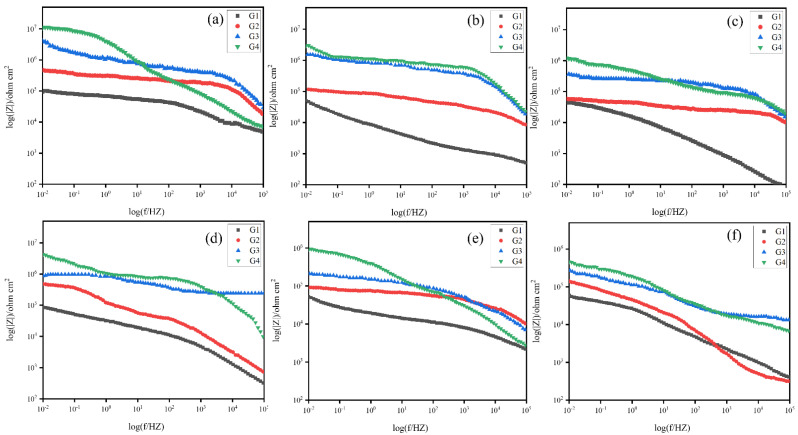
Bode diagram of all coatings served for 60 days in corrosive solutions of different concentrations: (**a**–**c**): After 45 days of service; (**d**–**f**): After 60 days of service.

**Figure 12 polymers-15-02602-f012:**
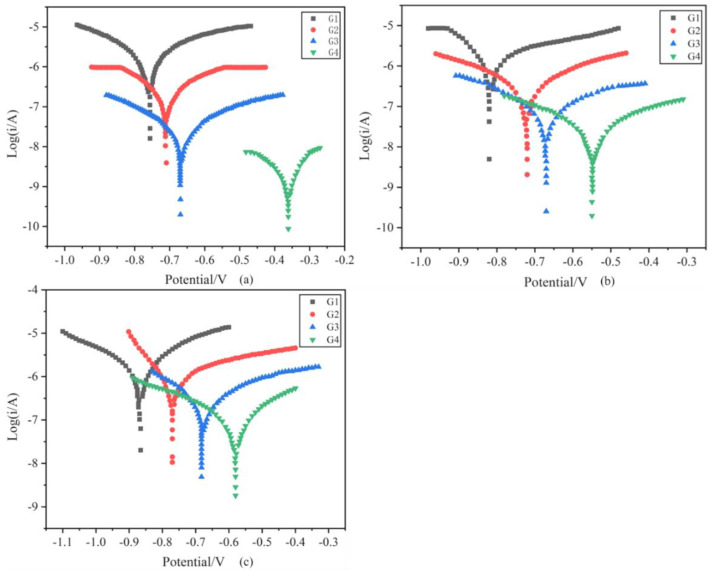
Tafel curve of each coating after 90 days of service: (**a**) 20% composite solution; (**b**) 40% composite solution; (**c**) 60% composite solution.

**Figure 13 polymers-15-02602-f013:**
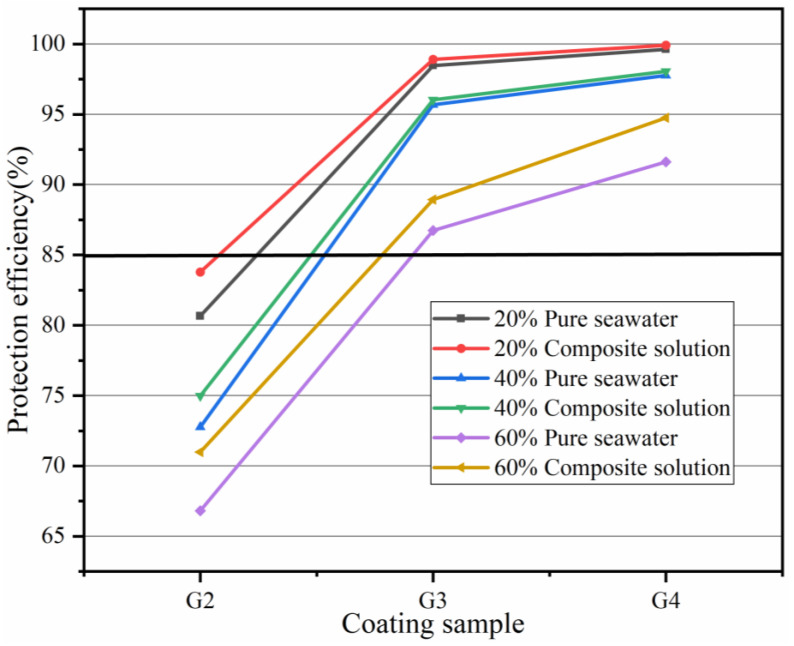
The protection efficiency of the coating after 90 days of service in different concentrations of solution.

**Figure 14 polymers-15-02602-f014:**
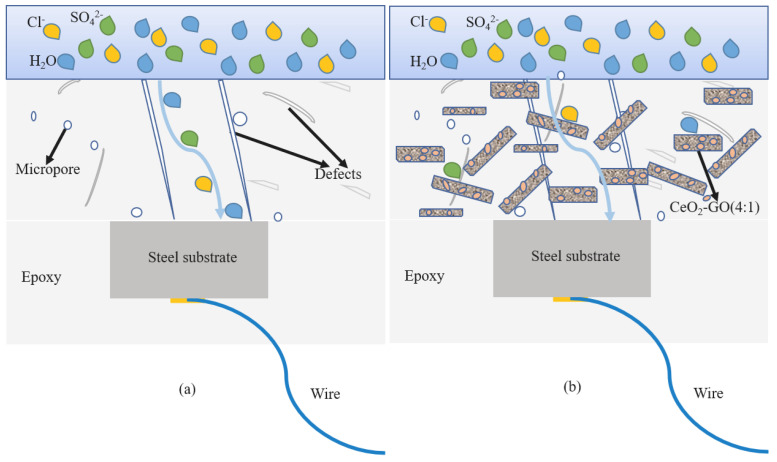
Schematic diagram of coating corrosion protection: (**a**) EP; (**b**) CeO_2_−GO/EP.

**Figure 15 polymers-15-02602-f015:**
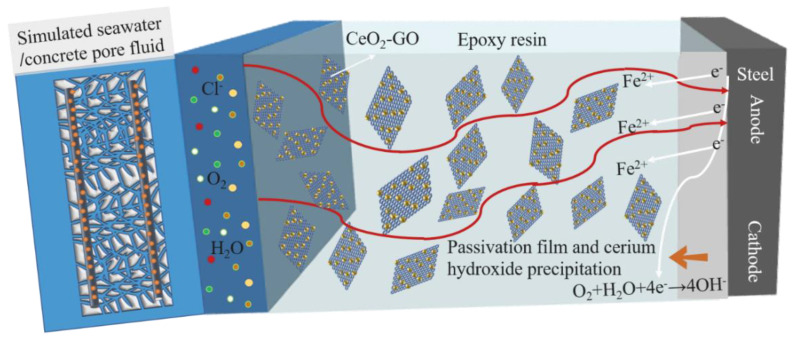
Schematic diagram of coating corrosion mechanism.

**Table 1 polymers-15-02602-t001:** Chemical composition of Q235 carbon steel.

Chemical Composition	C	Mn	Si	S	P	O	N	Fe
wt/(%)	0.15	0.68	0.12	0.026	0.023	0.01	0.003	98.988

**Table 2 polymers-15-02602-t002:** Basic performance of corrosion inhibitor.

Colour	pH Value	Density (g/cm^2^)	Corrosion Inhibition Rate
Light yellow to brownish red transparent liquid	9	1.030	>99.3%

**Table 3 polymers-15-02602-t003:** Comparison of coating samples.

Coating Sample				Composite Corrosion Solutions	Metal Substrate
G1	G2	G3	G4		
EP	EP+corrosion inhibitor	(CeO_2_-GO)/EP	(CeO_2_-GO)/EP+corrosion inhibitor	Simulated seawater and concrete pore liquid	Q235 low carbon steel

**Table 4 polymers-15-02602-t004:** Preparation of different concentration simulated seawater/concrete pore liquid.

Component	Concentration (g·L^−1^)		
	20%	40%	60%
SrCl_2_·6H_2_O	0.025	0.05	0.075
MgCl_2_·6H_2_O	5.20	10.4	15.6
CaCl_2_	1.16	2.32	3.48
NaF	0.003	0.006	0.009
KCl	0.695	1.39	2.085
NaHCO_3_	0.201	0.402	0.603
KBr	0.101	0.202	0.303
H_3_BO_3_	0.027	0.054	0.081
NaCl	24.53	24.53	24.53
Na_2_SO_4_	4.09	8.18	12.27
Ca(OH)_2_	Adjust the pH of the solution to 12.5		

**Table 5 polymers-15-02602-t005:** Adhesion evaluation method.

Adhesion Level/Level	Peeling Area (%)
0	none
1	(0, 5)
2	(5, 15)
3	(15, 35)
4	(35, 65)
5	>65

**Table 6 polymers-15-02602-t006:** Electrochemical testing projects and testing cycle.

Test Items	OCP	EIS	Tafel
Full testing cycle (d)	90	90	90

**Table 7 polymers-15-02602-t007:** Kinetic parameters of each coating after 90 days of service in 20% composite solution.

Coating	E_corr_ (V)	I_corr_ (A/cm^2^)	η (%)	R_p_ (ohm)
G1	−0.772	1.322 × 10^−6^	/	5.739 × 10^5^
G2	−0.703	2.143 × 10^−7^	83.79	7.663 × 10^6^
G3	−0.586	1.435 × 10^−8^	98.91	5.372 × 10^7^
G4	−0.364	1.001 × 10^−9^	99.92	9.473 × 10^8^

**Table 8 polymers-15-02602-t008:** Kinetic parameters of each coating after 90 days of service in 40% composite solution.

Coating	E_corr_ (V)	I_corr_ (A/cm^2^)	η (%)	R_p_ (ohm)
G1	−0.842	2.028 × 10^−6^	/	7.068 × 10^5^
G2	−0.729	5.072 × 10^−7^	74.99	2.300 × 10^6^
G3	−0.648	8.013 × 10^−8^	96.04	8.936 × 10^6^
G4	−0.483	3.942 × 10^−8^	98.06	3.110 × 10^7^

**Table 9 polymers-15-02602-t009:** Kinetic parameters of each coating after 90 days of service in 60% composite solution.

Coating	E_corr_ (V)	I_corr_ (A/cm^2^)	η (%)	R_p_ (ohm)
G1	−0.882	3.592 × 10^−6^	/	5.946 × 10^5^
G2	−0.764	1.042 × 10^−6^	70.99	8.583 × 10^5^
G3	−0.685	3.972 × 10^−7^	88.94	2.839 × 10^6^
G4	−0.573	1.882 × 10^−7^	94.76	6.029 × 10^6^

## Data Availability

Some or all data generated or used during the study are available from the corresponding author by request.

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
