# Peer review of "Corrosion Resistance of CeO2-GO/Epoxy Nanocomposite Coating in Simulated Seawater and Concrete Pore Solutions"

_polymers, 2023, doi:10.3390/polym15122602_

Round 1
Reviewer 1 Report
The manuscript reports on the morphological, mechanical, and electrochemical characteristics of epoxy coatings containing a CeO2-modified graphene oxide (GO) filler and pickling corrosion inhibitor, for the protection of construction carbon steel bars. In an incremental work, which is based on previous studies of epoxy-GO-CeO2 composite coatings, the authors present several interesting results regarding the mechanical and electrochemical performance of the coating after long-term exposure to simulated seawater and concrete pore solutions. Although these aspects might be of interest to the community, the study does not provide information on the structural properties of the filler in the epoxy matrix or evidence for the grafting of CeO2 on GO sheets. EDS, Raman, and XRD analyses are suggested. Furthermore, the work is poorly discussed and the suggested inhibiting mechanism is not supported by the presented data. In view of these points, some inconsistencies in data interpretation, and several other issues, listed below, the manuscript is not suited in the present form for publication in Polymers. Nevertheless, the following comments and suggestions could improve the manuscript for possible future publication.
Title suggestion:
Corrosion Resistance of CeO2-GO/Epoxy Nanocomposite Coating in Simulated Seawater and Concrete Pore Solutions
Abstract:
1- Reduce the introduction part and inform the Coating/Filler ratio, type of coated steel, and deposition method.
2- Revise the affirmation: “…his paper lays a solid theoretical foundation for the preparation…”
Introduction:
3- P. 5: Give more details on the “self-healing coordination mechanism”.
4- L. 51: Reconsider “…major challenges that the world is facing” in the phrase: ”Reinforcing bar corrosion …”
5- Check the validity of Refs. 15 and 16.
6- This is incorrect, please revise: “However, it is rare to study cerium oxide as a corrosion protection material in the field of corrosion”. There are numerous studies on CeO2 coatings and Ce(IV)/Ce(III) coating filers as corrosion inhibitors and self-healing agents.
7- L. 101: Discuss the challenges to coat 3D substrates such as steel reinforcing bars.
8- L. 104: Please clarify the novelty in this work in terms of refs 36-38.
9- L. 111-118: This is redundant information already presented in the Abstract.
10- L. 119: Section 2 is not part of the Introduction and is out of context. Move to Results and discussion.
Experimental:
11- For sake of clarity please rewrite the Experimental part. Avoid repetition, and join the information of electrochemical measurements.
12- L. 164: The used substrates on not represent the complex geometry of steel rebars. Therefore only the latter 3D geometry is adequate for the analysis of the anticorrosive properties.
13- L. 167: Detail on the coating process and the nature of the corrosion inhibitor.
14- L. 186: This is to clear: “The coating samples were evenly coated with (20±3) μm thickness on a carbon steel surface of 150 mm × 70 mm × 1mm, respectively.”
Which was the type of steel that was used and which was the coating thickness, 100 or 20 µm?
Results and discussion:
15- Discuss the results.
16- Fig. 3 is not clear, clarify this by extending the figure caption.
17- P. 7. Inform the type of hardness and adhesion measurements and the indentation depth.
18- L. 245: There is no evidence that the “Interface between the coating and the substrate is very rough”.
19- L. 251: There is no evidence for the affirmation: “The precipitation of cerium hydroxide generated by the hydrolysis of cerium oxide covers the surface…”
20- Fig. 5d: A uniform distribution of GO is not evident from the image, revise.
22- Fig. 6: Inform the Hardness unit [H]? Prefer using MPa.
23- L. 285: This affirmation is not evident from Fig. 7: “After 30 days of service, the adhesion grades of G1 and G2 decreased significantly…”. But it is clear that adhesion of G3 and G4 is inferior to pure EP, why?
24- Define “Adhesion grade” and include Unit in Fig. 7.
25- Fig. 9 -16: Present EIS results in a more compact form to make them comparable and include the carbons steel curves as reference. References for 1-day immersion could be also helpful.
26- L 409: Define “protection efficiency”
27- Conclusions: This phrase is not clear, please correct: “This paper restores the corrosion state of steel reinforcement applied in offshore buildings in different concentrations of the marine environment by adding simulated concrete pore liquid in simulated seawater.
28 L. 453: Clarify: “…due to the resolving effect of cerium ions…”
29- L. 456: This affirmation is not supported by the data: “The cerium oxide release into the corrosive solution hydrolyzes in the cathodic area of corrosion, generating insoluble cerium.” hydroxide precipitate (Ce 466 (OH)4);”
Minor points:
- Table 1. Correct G3 to CeO2-GO/EP and G4 to CeO2-GO/EP
- L. 240: Please check if “hybrid solution” is the adequate term.
- Fig. 4 and 5: Include in figure captions a description of images a) - d).
- Fig. 6 For clarity, use only 4 colors G1 - G4.
- Fig. 9 - 16: Include a) - c) in figures
Author Response
The manuscript reports on the morphological, mechanical, and electrochemical characteristics of epoxy coatings containing a CeO2-modified graphene oxide (GO) filler and pickling corrosion inhibitor, for the protection of construction carbon steel bars. In an incremental work, which is based on previous studies of epoxy-GO-CeO2 composite coatings, the authors present several interesting results regarding the mechanical and electrochemical performance of the coating after long-term exposure to simulated seawater and concrete pore solutions. Although these aspects might be of interest to the community, the study does not provide information on the structural properties of the filler in the epoxy matrix or evidence for the grafting of CeO2 on GO sheets. EDS, Raman, and XRD analyses are suggested. Furthermore, the work is poorly discussed and the suggested inhibiting mechanism is not supported by the presented data. In view of these points, some inconsistencies in data interpretation, and several other issues, listed below, the manuscript is not suited in the present form for publication in Polymers. Nevertheless, the following comments and suggestions could improve the manuscript for possible future publication.
Response:
Dear reviewer:
Thank you very much for taking your time to review this manuscript. We feel great thanks for your professional review work on our article. As you are concerned, there are many problems that need to be addressed. According to your nice suggestions, we have made extensive corrections to our previous draft, the detailed corrections are listed below. The revised paragraphs based on your comments in the manuscript are highlighted in red paragraphs.
Title suggestion:
Corrosion Resistance of CeO2-GO/Epoxy Nanocomposite Coating in Simulated Seawater and Concrete Pore Solutions
Response: Thank you very much for your professional guidance. I strongly agree with your suggestion and have revised the title to 'Corrosion Resistance of CeO2-GO/Epoxy Nanocomposite Coating in Simulated Seawater and Concrete Pore Solutions'.
Abstract:
1- Reduce the introduction part and inform the Coating/Filler ratio, type of coated steel, and deposition method.
Response: Thank you very much for your suggestion and it has been very helpful in polishing the article. The background section of the abstract has been reduced and the coating/filler ratio, coating steel type, and deposition method have been stated.
2- Revise the affirmation: “…his paper lays a solid theoretical foundation for the preparation…”
Introduction:
Response: Thank you very much for your professional guidance. The sentence ' This paper lays a solid theoretical foundation for the preparation of nano-composite coating with self-repairing function and provides a reference for solving the corrosion problem of reinforcement in the marine environment.' has been modified to ' This study provides a theoretical foundation for solving the corrosion problem of Q235 low carbon steel in the marine environment.'.
3- P. 5: Give more details on the “self-healing coordination mechanism”.
Response: Thank you very much for your valuable suggestion. Section 3.2 has been modified and a detailed introduction to the "self repair coordination mechanism" has been added.
The specific introduction is as follows:
CeO2-GO has a unique "corrosion self-repair coordination mechanism" that enables it to partially repair corrosion by itself. This mechanism is based on the interaction between two key materials: cerium oxide and graphene oxide. Cerium oxide has excellent redox properties and oxidant activity. When it interacts with graphene oxide, the latter exhibits self-repair properties due to its inherent stability and the redox reactions that it can undergo. When CeO2-GO is corroded, cerium oxide accepts electrons from graphene oxide, reduces the corroded area, and forms cerium ions. At the same time, graphene oxide absorbs electrons from cerium oxide and repairs itself in place. In this way, CeO2-GO can partially repair itself when subjected to corrosion through the interaction between cerium ions and graphene oxide, thereby maintaining its anti-corrosion properties. This "corrosion self-repair coordination mechanism" provides more reliable and long-lasting protection for anti-corrosion materials, enabling them to be used in more severe environments.
4- L. 51: Reconsider “…major challenges that the world is facing” in the phrase: ”Reinforcing bar corrosion …”
Response: We apologize for not expressing ourselves clearly. We have now rephrased this sentence and made the following modifications:
Reinforcing bar corrosion in the marine environment is one of the major challenges and it is therefore of great significance to study its protection measures.
5- Check the validity of Refs. 15 and 16.
Response: Thank you very much for your valuable feedback. I think this is an excellent suggestion. Given the validity of the references, they have been deleted and updated.
6- This is incorrect, please revise: “However, it is rare to study cerium oxide as a corrosion protection material in the field of corrosion”. There are numerous studies on CeO2 coatings and Ce(IV)/Ce(III) coating filers as corrosion inhibitors and self-healing agents.
Response: Thank you very much for your criticism and correction. We apologize for any issues with our presentation. This sentence has been replaced with a new expression.
The specific replacement content is as follows:
However, in marine environments, the corrosion protection of cerium oxide as a anti-corrosion material against steel bars in concrete pore solutions is still rarely studied.
7- L. 101: Discuss the challenges to coat 3D substrates such as steel reinforcing bars.
Response: Thank you very much for your review. We are very sorry that we did not discuss the corrosion protection of 3D substrates such as steel bars in this article. Such research will be covered in our future work.
8- L. 104: Please clarify the novelty in this work in terms of refs 36-38.
Response: We apologize for not presenting this point in the article and have now added it to the introduction section.
The title of this paper is "Corrosion Resistance of CeO2-GO/Epoxy Nanocomposite Coating in Simulated Seawater and Concrete Pore Solutions". Compared to previous research, this study has the following innovations:
Different research objects: The research object of this study is the CeO2-GO/epoxy nanocomposite coating and its corrosion resistance in simulated seawater and concrete pore solutions, while previous research mainly focused on the corrosion resistance of CeO2-GO nanocomposites in different environments.
Changes in experimental conditions: The experimental conditions of this study include simulated seawater and concrete pore solutions, whose chemical composition and pH values are different from those used in previous research. Therefore, a new experimental plan and testing method need to be designed.
Expansion of application scenarios: This study involves concrete pore solutions. Concrete is a widely used building material, and its corrosion problems have an important impact on the safety and service life of building structures. Therefore, this study is of practical significance for the corrosion protection of concrete buildings.
Overall, the innovation of this study lies in the application of CeO2-GO nanocomposites to epoxy coatings and the exploration of their corrosion resistance in simulated seawater and concrete pore solutions, as well as the application prospects for corrosion protection of concrete structures. The significance of this work is to expand the application of this composite material and provide a new solution for the corrosion protection of concrete buildings.
9- L. 111-118: This is redundant information already presented in the Abstract.
Response: Thank you very much for your guidance, I completely agree. This section has now been deleted.
10- L. 119: Section 2 is not part of the Introduction and is out of context. Move to Results and discussion.
Response: Thank you very much for your comments and guidance. We apologize for mistakenly placing this section. The second part has been moved to section 3.5 of the Results and Discussion section.
Experimental:
11- For sake of clarity please rewrite the Experimental part. Avoid repetition, and join the information of electrochemical measurements.
Response: Thank you very much for your revision comments. It has been helpful in organizing the logic of the article. The experimental section of the article has been revised and electrochemical measurement information has been added. The experimental section has been modified as follows:
- Experimental
2.1. Raw materials
Graphene oxide was supplied by Changzhou Sixth Element Materials Technology Co., Changzhou, China. Cerium nitrate hexahydrate was purchased from Shanghai Aladdin Biochemical Technology Co., Shanghai, China. Epoxy resin (WSR6101 E-44) and epoxy AB adhesive were supplied by Nantong Xingchen Synthetic Materials Co, Nantong, China. The epoxy resin (WSR6101 E-44) is a bisphenol thermosetting epoxy resin with a viscosity of 15000 mPa·s at 25 °C. The selected Lanxing Lan-826 multifunctional pickling corrosion inhibitor was purchased from Henan Xinyang Chemical Co, Henan Province, China. CaCl2, NaF, KCl, NaHCO3, KBr, H3BO3, NaCl, and Na2SO4 were supplied by Chengdu Kolon Chemical Co., Chengdu, China. Table 1 describes the chemical composition of Q235 steel sheet. The basic performance of Lan-826 corrosion inhibitor is shown in Table 2.
Table 1 Chemical composition of Q235 carbon steel
|
Chemical composition |
C |
Mn |
Si |
S |
P |
O |
N |
Fe |
|
wt/(%) |
0.15 |
0.68 |
0.12 |
0.026 |
0.023 |
0.01 |
0.003 |
98.988 |
Table 2 Basic performance of corrosion inhibitor
|
Colour |
pH value |
Density (g/cm2) |
Corrosion inhibition rate |
|
Light yellow to brownish red transparent liquid |
9 |
1.030 |
>99.3% |
2.2. Preparation of CeO2-GO (4:1) / EP nanocomposite coating
Dissolve 10g of sodium bicarbonate and 186.6mg of ethylenediaminetetraacetic acid disodium salt dihydrate (EDTA.2Na) in 500ml of deionized water, stir well, and place in a dialysis bag. Boil the solution in the dialysis bag for 10 minutes. After cleaning the dialysis bag, add the graphene oxide liquid and boil again in a beaker for 10 minutes. Then, dissolve the liquid in deionized water, disperse it by ultrasonic treatment for 30 minutes, transfer it into the dialysis bag, and stir it in a beaker for 3 days. Finally, extract the liquid from the dialysis bag and use an ultrasonic cell disrupter to obtain a graphene oxide dispersion. Weigh 0.807g of cerium nitrate hexahydrate and dissolve it in a beaker with 4ml of ammonia solution. Stir the solution with a glass rod and then transfer it into a polytetrafluoroethylene-lined container, which is placed inside a high-pressure reactor. Put the reactor into an oven set at 160℃ for 24 hours. After the reaction is complete, filter, dry, and grind the product to obtain CeO2-GO nanoparticles.
CeO2-GO has a unique "corrosion self-repair coordination mechanism" that enables it to partially repair corrosion by itself. This mechanism is based on the interaction between two key materials: cerium oxide and graphene oxide. Cerium oxide has excellent redox properties and oxidant activity. When it interacts with graphene oxide, the latter exhibits self-repair properties due to its inherent stability and the redox reactions that it can undergo. When CeO2-GO is corroded, cerium oxide accepts electrons from graphene oxide, reduces the corroded area, and forms cerium ions. At the same time, graphene oxide absorbs electrons from cerium oxide and repairs itself in place. In this way, CeO2-GO can partially repair itself when subjected to corrosion through the interaction between cerium ions and graphene oxide, thereby maintaining its anti-corrosion properties. This "corrosion self-repair coordination mechanism" provides more reliable and long-lasting protection for anti-corrosion materials, enabling them to be used in more severe environments. In the experiments, Q235 steel sheets(150 mm × 70 mm × 1 mm) were used to assess the hardness and adhesion of the coating, and Q235 mild steel with dimensions of 5mm in height and 10mm in diameter was selected to evaluate the anticorrosive properties of the coating. The prepared CeO2-GO composites were mixed with epoxy resin at a mass ratio of 0.5% and coated on the bottom of the electrode. After the coating solidified, it was rubbed with the sandpaper to ensure that the coating thickness is controlled at 100 μm by a coating thickness gauge. The samples were immersed in simulated seawater and concrete pore liquid composite solutions for 90 days to test the coating's corrosion resistance.
For comparison, as shown in Table 2, four epoxy coating samples, G1, G2, G3, and G4, were prepared with G1 being the pure epoxy coating, G2 being epoxy coating with corrosion inhibitor, G3 being CeO2-GO/EP coating, G4 being CeO2-GO/EP coating with corrosion inhibitor. The anti-corrosion performance of the four coatings subjected to simulated seawater and simulated concrete pore liquid composite solutions were evaluated.
Table 2. Comparison of coating samples.
|
Coating sample |
Composite corrosion solutions |
Metal substrate |
|||
|
G1 |
G2 |
G3 |
G4 |
||
|
EP |
EP+corrosion inhibitor |
(CeO2-GO)/EP |
(CeO2-GO)/EP+corros-ion inhibitor |
Simulated seawater and concrete pore liquid |
Q235 low carbon steel |
2.3. Preparation of corrosive solutions
Different concentrations of simulated seawater and simulated concrete pore liquid composite solutions (20%, 40%, 60%) were prepared according to ASTM D1141-1998 (2013). As shown in Figure 1, 24.53 g of NaCl and 4.09 g of Na2SO4 were dissolved in 800 ml of the aqueous solution. Slowly add 20 mL of stock solution No. 1 with vigorous stirring, then add 10 mL of stock solution No. 2. The solution pH was adjusted to 12.5 with Ca(OH)2 after dilution to 1 L. Table 3 shows the chemical composition of the simulated seawater and simulated concrete pore solution.
Figure 1. Schematic diagram of preparation of corrosive solution: preparation of corrosive solution and optical microscope morphology after corrosion.
Table 3. Preparation of different concentration simulated seawater/concrete pore liquid.
|
Component |
Concentration(g·L-1) |
||
|
20% |
40% |
60% |
|
|
SrCl2·6H2O |
0.025 |
0.05 |
0.075 |
|
MgCl2·6H2O |
5.20 |
10.4 |
15.6 |
|
CaCl2 |
1.16 |
2.32 |
3.48 |
|
NaF |
0.003 |
0.006 |
0.009 |
|
KCl |
0.695 |
1.39 |
2.085 |
|
NaHCO3 |
0.201 |
0.402 |
0.603 |
|
KBr |
0.101 |
0.202 |
0.303 |
|
H3BO3 |
0.027 |
0.054 |
0.081 |
|
NaCl |
24.53 |
24.53 |
24.53 |
|
Na2SO4 |
4.09 |
8.18 |
12.27 |
|
Ca(OH)2 |
Adjust the pH of the solution to 12.5 |
||
2.4. Test method
2.4.1. Corrosion micromorphology testing
To visualize the microstructure of CeO2-GO nanocomposite coatings in the seawater and concrete pore liquid environments with long-term service, a scanning electron microscope (FESEM, Gemini, M/s.Zeiss, Germany) was used with a working distance of 7 mm and an acceleration voltage of 1 kV at room temperature. With pure epoxy coating as a contrast, G1 and G3 samples were immersed in 60% hybrid solution for 60 days to characterize the micromorphology of the corroded coating substrate and thus to evaluate the anti-corrosion effect of CeO2-GO (4:1) nanocomposites.
2.4.2. Hardness test
The hardness of the coatings was tested according to standard ISO 15184:1998 [37]. The hardness of a coating was determined using the pencil scratch hardness test. The hardness of the hardest pencil that does not leave a scratch exceeding 3 millimeters on the coating is used to represent the hardness of the coating. The higher the hardness of the coating, the better the wear resistance of the coating. The coating samples were evenly coated with (20±3) μm thickness on a Q235 steel surface of 150 mm × 70 mm × 1mm, respectively. After the samples were cured at room temperature for 24h, the hardness of the coatings were measured at a temperature of (23±2)℃ and relative humidity of (50±5)%. All samples were immersed in different concentrations of the solutions simultaneously for 15 days to obtain the hardness of the coatings after service in the simulated seawater and simulated concrete pore liquid composite solutions.2.4.3. Adhesion Test
The adhesion values of the coatings were determined according to standard ISO 2409:2013 [38]. The sample preparation is the same as the hardness test. The coating samples were evenly coated with (20±3) μm thickness on a carbon steel surface of 150 mm × 70 mm × 1mm, respectively. After each sample is immersed in the solution for 30 days, the coating adhesion tester is used to conduct the test under the conditions of temperature (23 ± 2) ℃ and relative humidity (50 ± 5)%. First, the cured coating template is fixed in a position perpendicular to a grid device, and a cross-shaped scratch is made on it. Then, a transparent tape is uniformly covered over the scratch. Finally, the tape is removed, and the adhesion strength is determined by observing whether the scratch comes off or not according to certain criteria. As shown in Table 4, the adhesion grade is divided into six grades according to the area of the coating surface peeling off. The smaller the grade, the smaller the area of the coating peeling off and the better the adhesion.
Table 4. Adhesion evaluation method
|
Adhesion level/level |
Peeling area(%) |
|
0 |
none |
|
1 |
(0,5) |
|
2 |
(5,15) |
|
3 |
(15,35) |
|
4 |
(35,65) |
|
5 |
>65 |
2.4.4. Electrochemical test
Electrochemical tests were performed using a CHI-760E electrochemical workstation with a three-electrode system. The platinum electrode and saturated glycury electrode were used as counter electrode and reference electrode, respectively. G1, G2, G3, and G4 were conducted as working electrodes after the open circuit potential values were stabilized. In electrochemical experiments, the thickness of all coatings is controlled at 100 μm. In EIS testing, The initial potential (Init E (V)) is the stable value of the open circuit potential, the high frequency (Hz) is 100000, the low frequency (Hz) is 0.01, the amplitude (V) is 0.005, and the quiet time (sec) is 2.
After the samples were immersed in the composite solution for 90 days, the tafel polarization curves were recorded at a scanning speed of 0.5 mV/s in the range of -200 mV to +1200 mV to investigate the trends of kinetic parameters of the coatings after long-term service in simulated seawater and simulated concrete pore liquid composite solutions with different concentrations. The table 5 describes the time when all electrochemical tests were done. The EIS and Tafel tests were completed simultaneously within 90 days. In the EIS testing, as there was no significant difference between the 90 day test results and the 60 day test results, in order to avoid data redundancy, section 4.4.2 only analyzed the EIS test results within 60 days.
Table 5. Electrochemical testing projects and testing cycle
|
Test items |
OCP |
EIS |
Tafel |
|
Full testing cycle (d) |
90 |
90 |
90 |
12- L. 164: The used substrates on not represent the complex geometry of steel rebars. Therefore only the latter 3D geometry is adequate for the analysis of the anticorrosive properties.
Response: I completely agree with your viewpoint. The substrate used above is used to illustrate the schematic diagram of the electrochemical sample. The latter 3D geometric shape is used to analyze anti-corrosion performance.
13- L. 167: Detail on the coating process and the nature of the corrosion inhibitor.
Response: Detailed information on the coating process has been added to section 2.2. In addition, table 2 has been added to describe the properties of corrosion inhibitors. (Please refer to page 6)
Table 2 Basic performance of corrosion inhibitor
|
Colour |
pH value |
Density (g/cm2) |
Corrosion inhibition rate |
|
|
Light yellow to brownish red transparent liquid |
9 |
1.030 |
>99.3% |
14- L. 186: This is to clear: “The coating samples were evenly coated with (20±3) μm thickness on a carbon steel surface of 150 mm × 70 mm × 1mm, respectively.”
Which was the type of steel that was used and which was the coating thickness, 100 or 20 µm?
Response: Thank you very much for your modification suggestions. The coating thickness for hardness and adhesion experiments is 20 μm. Electrochemical test coating thickness is 100 μm. The coating thicknesses for all tests are specified in sections 2.4.2, 2.4.3, and 2.4.4, respectively. The Q235 steel was used, and the elemental composition of the steel used in the experiment has been described in Table 1 of Section 2.1.
Results and discussion:
15- Discuss the results.
Response: Thank you very much for your professional guidance. In order to discuss the results, section 3.5 has been added.
16- Fig. 3 is not clear, clarify this by extending the figure caption.
Response: We apologize for the lack of clarity in original Figure 3.and we have now clarified this point by extending the icon question. The changes made are shown below.
Figure 1. Schematic diagram of preparation of corrosive solution: preparation of corrosive solution and optical microscope morphology after corrosion.
17- P. 7. Inform the type of hardness and adhesion measurements and the indentation depth.
Response:The hardness of the coatings was tested according to standard ISO 15184:1998. The hardness of a coating was determined using the pencil scratch hardness test. The hardness of the hardest pencil that does not leave a scratch exceeding 3 millimeters on the coating is used to represent the hardness of the coating. The adhesion values of the coatings were determined according to standard ISO 2409:2013. First, the cured coating template is fixed in a position perpendicular to a grid device, and a cross-shaped scratch is made on it. Then, a transparent tape is uniformly covered over the scratch. Finally, the tape is removed, and the adhesion strength is determined by observing whether the scratch comes off or not according to certain criteria. The adhesion grade is divided into six grades according to the area of the coating surface peeling off. The smaller the grade, the smaller the area of the coating peeling off and the better the adhesion.
18- L. 245: There is no evidence that the “Interface between the coating and the substrate is very rough”.
Response: Thank you very much for your modification suggestions. I'm sorry that this statement is not very appropriate. Through SEM morphology analysis, it was found that the pure EP coating severely corroded after corrosion, with deep corrosion pits and extensive peeling of the coating. The corrosion degree of CeO2-GO (4:1) coating significantly decreased. The sentence "the interface between the coating and the substrate is very rough " has been revised to "The pure EP coating has a severe degree of corrosion, and there are deep corrosion pits due to large-scale peeling of the coating.”
19- L. 251: There is no evidence for the affirmation: “The precipitation of cerium hydroxide generated by the hydrolysis of cerium oxide covers the surface…”
Response: Thank you very much for your guidance. In our previous work, we conducted SEM-EDS scanning tests on CeO2-GO (4:1), which showed that CeO2 was successfully grafted onto the surface of GO. Therefore,the phrase 'cerium hydroxide precipitate generated by cerium oxide hydrolysis covers the surface' is a speculation on the corrosion mechanism, and We apologize for not being suitable for expressing it in this way. It has now been deleted.
20- Fig. 5d: A uniform distribution of GO is not evident from the image, revise.
Response: Thank you very much for your comment. Figure 5(d) has been corrected.
22- Fig. 6: Inform the Hardness unit [H]? Prefer using MPa.
Response: Thank you very much for your suggestion. The hardness of the coatings was tested according to standard ISO 15184:1998. The hardness of a coating was determined using the pencil scratch hardness test. The hardness of the hardest pencil that does not leave a scratch exceeding 3 millimeters on the coating is used to represent the hardness of the coating. Therefore, [H] was used as the unit of hardness in the experiment. In future research on coating hardness, we will prioritize using Mpa as the unit of hardness using other methods.
23- L. 285: This affirmation is not evident from Fig. 7: “After 30 days of service, the adhesion grades of G1 and G2 decreased significantly…”. But it is clear that adhesion of G3 and G4 is inferior to pure EP, why?
Response: I'm sorry for not explaining it clearly in the text. The adhesion level evaluation method has now been defined in section 2.4.3. The adhesion grade is divided into six grades according to the area of the coating sur-face peeling off. The smaller the level, the smaller the area of the coating peeling off and the better the adhesion. After 30 days of use. The peeling area of G3 and G4 is smaller, and according to the classification of adhesion level, their adhesion level is lower compared to G1 and G2. Therefore, after 30 days of service, the adhesion of G3 and G4 is due to pure EP.
Table 4. Adhesion evaluation method
|
Adhesion level/level |
Peeling area(%) |
|
0 |
none |
|
1 |
(0,5) |
|
2 |
(5,15) |
|
3 |
(15,35) |
|
4 |
(35,65) |
|
5 |
>65 |
24- Define “Adhesion grade” and include Unit in Fig. 7.
Response: Thank you for your professional guidance. Adhesion testing method has been added in section 2.4.3, and a table has been added to illustrate the basis for adhesion evaluation.
25- Fig. 9 -16: Present EIS results in a more compact form to make them comparable and include the carbons steel curves as reference. References for 1-day immersion could be also helpful.
Response: Thank you very much for your valuable feedback. In order to present clear and comparable images, the EIS results have been modified to be presented in a more compact manner in the text. Additionally, I fully agree with your point of view that in future research, 1-day corrosion results including low-carbon steel will be considered.
26- L 409: Define “protection efficiency”
Response: Protection efficiency has been defined in section 3.4.3.
27- Conclusions: This phrase is not clear, please correct: “This paper restores the corrosion state of steel reinforcement applied in offshore buildings in different concentrations of the marine environment by adding simulated concrete pore liquid in simulated seawater.
Response: I'm sorry for not expressing myself clearly. This sentence has been revised to “This paper investigates the corrosion behavior of steel reinforcement in offshore structures under different concentrations of marine environments. The study involves introducing simulated concrete pore liquid into simulated seawater to simulate the actual corrosive conditions.”
28 L. 453: Clarify: “…due to the resolving effect of cerium ions…”
Response: I'm sorry for not explaining it clearly in the text. Due to our previous work on SEM-EDS testing of CeO2-GO (4:1), we found that cerium oxide was successfully synthesized on the surface of GO. Based on the results of electrochemical experiments, we speculate that the hydrolysis product of cerium oxide inhibits metal corrosion. This sentence has now been corrected in the text.
29- L. 456: This affirmation is not supported by the data: “The cerium oxide release into the corrosive solution hydrolyzes in the cathodic area of corrosion, generating insoluble cerium.” hydroxide precipitate (Ce 466 (OH)4);”
Response: We have already conducted SEM-EDS experiments on CeO2-GO (4:1) in our previous work, and we apologize for not explaining it in the text. Now it has been corrected and the SEM-EDS test results for CeO2-GO (4:1) have been added again in section 3.1.
Minor points:
- Table 1. Correct G3 to CeO2-GO/EP and G4 to CeO2-GO/EP
Response: Thank you for your professional guidance. In Table 1, G3 has been corrected to CeO2-GO/EP, and G4 has been corrected to CeO2-GO/EP+corrosion inhibitor.
Table 2. Comparison of coating samples.
|
Coating sample |
Composite corrosion solutions |
Metal substrate |
|||
|
G1 |
G2 |
G3 |
G4 |
||
|
EP |
EP+corrosion inhibitor |
(CeO2-GO)/EP |
(CeO2-GO)/EP+corros-ion inhibitor |
Simulated seawater and concrete pore liquid |
Q235 low carbon steel |
- L. 240: Please check if “hybrid solution” is the adequate term.
Response: Thank you very much for your professional guidance. The term 'hybrid solution' has been modified to 'corrosive solution'.
- Fig. 4 and 5: Include in figure captions a description of images a) - d).
Response: I highly appreciate your suggestion. In original Figures 4 and 5, descriptions of images a) - d) have been added to the image titles. The specific modifications are as follows:
Figure 2. SEM images of EP in 60% corrosive solution for 60 days: (a) corroded surface; (b) corrosion products; (c) locally corroded; (d) corrosion amplification area; (e) optical microscopic morphology after 60 days of corrosion.
Figure 3. SEM images of CeO2-GO/EP in 60% corrosive solution for 60 days: (a) corroded surface; (b) corrosion products; (c) locally corroded; (d) corrosion amplification area; (e) optical microscopic morphology after 60 days of corrosion.
- Fig. 6 For clarity, use only 4 colors G1 - G4.
Response: Thank you very much for your suggestion, which has greatly helped to improve the quality of the article. For clarity, Figure 6 in the original text has been modified and marked G1-G4 with four colors. After modification, it is shown in the following figure.
- Fig. 9 - 16: Include a) - c) in figures
Response:Thank you for your guidance. (a), (b), and (c) in Figures 9-16 are marked below each small image.

Reviewer 2 Report
The manuscript presents an interesting study regarding the corrosion behavior of steel coated with a CeO2-GO/EP nanocomposite layer in two different corrosive media: simulated seawater and simulated concrete liquid composite solution. The samples were also characterized from the point of view of hardness and adhesion. However, the paper needs major revisions before it is processed further. Some comments follow:
Abstract
The abstract is too long; the background and conclusion parts must be reduced.
Experimental
Subsection 3.1: Introduce the information regarding the used steel. Also, introduce a table with its chemical composition.
Subsection 3.3: Please replace the word "composite" with "corrosive" since just one of them is a composite solution. Also, in the figure 3 title, replace "corrosion" with "corrosive".
Lines 186-196. This paragraph must be moved into the 3.4.4 subsection or removed since it contains almost the same information. Also, it is necessary to add the exposed surface of the working electrode. Please add information regarding the software used for EIS results.
It is not clear at what time all the electrochemical tests were done. Please add a table where you can put this information. Probably it is a reason why the authors chose to make the other test last 60 days. Please also write down this information. Since the only test at 90 days is the PDP test.
Results and discussion
Lines 247-248. How was the erosion portion of the samples calculated? Please introduce this information.
It is not enough to introduce and discuss just the Nyquist and Bode diagrams; the authors must also add the equivalent circuits for all the situations. A table with the parameter values of the equivalent circuits must be added and discussed also. If you consider the Bode and Nyquist diagrams, they can be introduced in the Appendix.
Subsection 4.4.3: The Rp is calculated taking into consideration the values of a and c; please introduce these values. Also, it will be interesting to calculate the corrosion rate, introduce those values, and discuss them.
Conclusions
The conclusion section is too long. Please introduce the most important of them, as well as some quantitative results.
References
Are too many self-citations. Please remove the unnecessary ones.
Author Response
The manuscript presents an interesting study regarding the corrosion behavior of steel coated with a CeO2-GO/EP nanocomposite layer in two different corrosive media: simulated seawater and simulated concrete liquid composite solution. The samples were also characterized from the point of view of hardness and adhesion. However, the paper needs major revisions before it is processed further. Some comments follow:
Response:
Dear reviewer:
Thank you for taking your time to review our manuscript. We appreciate your constructive comments and suggestions. We are grateful for the opportunity to address the issues you raised and improve the quality of our work. We agree that our manuscript needs significant revision before it can be further processed. We will carefully consider your feedback and make the necessary changes to improve the clarity and scientific rigor of our research. The revised paragraphs based on your comments in the manuscript are highlighted in yellow.
Abstract
The abstract is too long; the background and conclusion parts must be reduced.
Response: Thank you very much for your valuable suggestion. The abstract has been reduced, and unnecessary parts of the background and conclusion have been removed. Now, the summary is modified as follows:
Reinforced concrete structures in the marine environment face serious corrosion risks. Coating protection and adding corrosion inhibitors are the most economical and effective methods. In this study, a nano-composite anti-corrosion filler with a mass ratio of CeO2:GO = 4:1 was prepared by hydrothermally growing cerium oxide on the surface of graphene oxide. The filler was mixed with pure epoxy resin at a mass fraction of 0.5% to prepare a nano-composite epoxy coating. The basic properties of the prepared coating were evaluated from the aspects of surface hardness, adhesion grade, and anti-corrosion performance on Q235 low carbon steel subjected to simulated seawater and simulated concrete pore solutions. Results showed that after 90 days of service, the corrosion current density of the nanocomposite coating mixed with corrosion inhibitor was the lowest (Icorr = 1.001×10-9A/cm2), and the protection efficiency was up to 99.92%. This study provides a theoretical foundation for solving the corrosion problem of Q235 low carbon steel in the marine environment.
Experimental
Subsection 3.1: Introduce the information regarding the used steel. Also, introduce a table with its chemical composition.
Response: Thank you very much for your valuable suggestion. The information on the steel used has been introduced in section 2.1. A table with its chemical composition has also been presented in section 2.1. The modifications made are as follows:
Table 1 Chemical composition of Q235 carbon steel
|
Chemical composition |
C |
Mn |
Si |
S |
P |
O |
N |
Fe |
|
wt/(%) |
0.15 |
0.68 |
0.12 |
0.026 |
0.023 |
0.01 |
0.003 |
98.988 |
Subsection 3.3: Please replace the word "composite" with "corrosive" since just one of them is a composite solution. Also, in the figure 3 title, replace "corrosion" with "corrosive".
Response: Thank you very much for your valuable suggestion. In original section 3.3, the term "composite solution" has been changed to "corrosive solution". In the title of Figure 3, 'corrosion' has been replaced with 'corrosive'.
Lines 186-196. This paragraph must be moved into the 3.4.4 subsection or removed since it contains almost the same information. Also, it is necessary to add the exposed surface of the working electrode. Please add information regarding the software used for EIS results.
Response: Thank you very much for your valuable suggestion. Lines 186-196. This paragraph has been deleted and the relevant content has been moved to 2.4.2, 2.4.3, and 2.4.4, respectively. In addition, the exposed surface of the working electrode has been added to section 2.4.4. The information on the software used for EIS results has also been supplemented in section 2.4.4.
It is not clear at what time all the electrochemical tests were done. Please add a table where you can put this information. Probably it is a reason why the authors chose to make the other test last 60 days. Please also write down this information. Since the only test at 90 days is the PDP test.
Response: Thank you very much for your valuable suggestion. In section 3.4.4, a table has been added to describe the completion time of all electrochemical tests. The EIS and Tafel tests were completed simultaneously within 90 days. In the EIS testing, as there was no significant difference between the 90 day test results and the 60 day test results, in order to avoid data redundancy, section 4.4.2 only analyzed the EIS test results within 60 days. We can provide the EIS test results for 90d if needed. I'm sorry for not explaining the reason in the text due to our negligence.
Table 5. Electrochemical testing projects and testing cycle
|
Test items |
OCP |
EIS |
Tafel |
|
Full testing cycle (d) |
90 |
90 |
90 |
Results and discussion
Lines 247-248. How was the erosion portion of the samples calculated? Please introduce this information.
Response: Thank you very much for your professional guidance. I think this is a very valuable suggestion. Calculate the corrosion on the surface of the sample using an optical microscope. The specific calculation method is as follows: select the observation area and determine its size and shape. The size of the observed area can be measured using the eyepiece scale and objective scale of a microscope. Based on the measured shape and size, the area of the corroded part of the sample can be calculated. This information has now been supplemented in section 2.4.1.
It is not enough to introduce and discuss just the Nyquist and Bode diagrams; the authors must also add the equivalent circuits for all the situations. A table with the parameter values of the equivalent circuits must be added and discussed also. If you consider the Bode and Nyquist diagrams, they can be introduced in the Appendix.
Response: Thank you very much for your professional guidance. This article mainly explores its corrosion resistance and mechanical properties in simulated seawater and concrete pore solutions, as well as its application prospects for corrosion protection of concrete structures. We apologize for not adding an equivalent circuit in this article. Your suggestion is very valuable. In future research, we will add a table containing equivalent circuit parameter values and discuss it. Thank you.
Subsection 4.4.3: The Rp is calculated taking into consideration the values of a and c; please introduce these values. Also, it will be interesting to calculate the corrosion rate, introduce those values, and discuss them.
Response: Thank you very much for your valuable suggestion. The Tafel curve represents the strongly polarized part of the polarization curve. By studying the relationship between corrosion current density and corrosion voltage, the damage rate and protection mechanism of anti-corrosion coatings are analyzed. When calculating Rp, the values of ba and bc were taken into account, and the relevant introduction has been added to section 4.4.3. Currently, the main research focus is on the polarization resistance and protection efficiency of coatings. I strongly agree with your opinion and it is very helpful for improving the quality of the article. The corrosion rate will be studied in subsequent corrosion studies. The specific modifications are as follows:
The Tafel curve represents the strongly polarized part of the polarization curve. By studying the relationship between corrosion current density and corrosion voltage, the polarization resistance and protection efficiency of anti-corrosion coatings are analyzed. The expression is as follows.
(1)
represents the slope of the Tafel curve of the electrode anode. represents the slope of the Tafel curve of the electrode cathode. represents the current corrosion density of the anti-corrosion coating. represents the current corrosion density of a pure epoxy resin coating. is the coating polarization resistance of the prepared electrode. represents the coating protection efficiency.
Conclusions
The conclusion section is too long. Please introduce the most important of them, as well as some quantitative results.
Response: Thank you very much for your guidance. I completely agree with your viewpoint. The conclusion section has been reduced. The specific modifications regarding the conclusion in the fourth part are as follows:
In this paper, CeO2-GO (4:1) nanocomposites were produced by hydrothermal synthesis, and CeO2-GO/EP coatings were formulated by mixing with epoxy resin. This paper investigates the corrosion behavior of steel reinforcement in offshore structures under different concentrations of marine environments. The study involves introducing simulated concrete pore liquid into simulated seawater to simulate the actual corrosive conditions. The corrosion resistance and mechanical properties of CeO2-GO/EP in simulated seawater/concrete pore liquid composite solutions with different concentrations were investigated. The corrosion areas of the coatings after 60 d of service were calculated by optical microscopy, and the corrosion area of CeO2-GO (4:1)/EP was reduced by half compared to that of pure epoxy (41.4%), with a value of 22.8%. The evolution of the corrosion state of the coatings was investigated by electrochemical tests during 90d , indicating that the low-frequency impedance value of CeO2-GO (4:1)/EP doped with corrosion inhibitor was the largest (|Z|0.01 = 108 ohm cm2). By calculating Tafe polarization data, the corrosion current density of CeO2 GO (4:1)/EP after 90 days of service is Icorr=1.435 × 10-8A/cm2. After calculation and analysis, the protection efficiency of CeO2-GO (4:1)/EP is 98.91%, significantly improving the corrosion resistance of the coating.
References
Are too many self-citations. Please remove the unnecessary ones.
Response: Thank you very much for your valuable suggestion. We fully agree with your suggestion. Unnecessary references have been removed. Please refer to page 31 for specific modification results.

Round 2
Reviewer 1 Report
The manuscript reports on the morphological, mechanical, and electrochemical characteristics of epoxy coatings containing a CeO2-modified graphene oxide (GO) filler and pickling corrosion inhibitor, for the protection of construction carbon steel bars. The authors have responded satisfactorily the most part of the reviewers' comments and suggestions, however, there are some issues that should be addressed for the final version of the manuscript.
Regarding pints:
-3- P. 5: (Give more details on the “self-healing coordination mechanism”.)
“When CeO2-GO is corroded, cerium oxide accepts electrons from graphene oxide, reduces the corroded area, and forms cerium ions.” This phrase is not clear (…CeO2-GO is corroded …, cerium oxide (CeO2) accepts electrons …, and forms cerium ions?). Please revise and provide references.
-11- (For the sake of clarity please rewrite the Experimental part. Avoid repetition, and join the information of electrochemical measurements.)
Please write the experimental part in the passive form and separate for quantities unit from the numerical value: 10 g of sodium bicarbonate and 186.6 mg… was dissolved…
Move the paragraph in 2.2: “CeO2-GO has a unique corrosion self-repair coordination mechanism that enables it to partially repair…” into Results and Discussion.
-13- L. 167: (Detail on the coating process …).
Inform the type of coating process.
- 15- (Discuss the results.) Include references in the Results and discussion and in section 3.5.
- 25- (Fig. 9 -16: Present EIS results in a more compact form to make them comparable). There are 6 figures of EIS results. Reduce the number by joining Bode Plots. An EIS reference of bare carbon steel is necessary to evaluate the protection efficiency of the coatings.
Author Response
The manuscript reports on the morphological, mechanical, and electrochemical characteristics of epoxy coatings containing a CeO2-modified graphene oxide (GO) filler and pickling corrosion inhibitor, for the protection of construction carbon steel bars. The authors have responded satisfactorily the most part of the reviewers' comments and suggestions, however, there are some issues that should be addressed for the final version of the manuscript.
Response:
Dear reviewer:
Thank you for taking your time to review our manuscript. We appreciate your constructive comments and suggestions. We are grateful for the opportunity to address the issues you raised and improve the quality of our work. We will carefully consider your feedback and make the necessary changes to improve the clarity and scientific rigor of our research. The revised paragraphs based on your comments in the manuscript are highlighted in red.
Regarding pints:
-3- P. 5: (Give more details on the “self-healing coordination mechanism”.)
“When CeO2-GO is corroded, cerium oxide accepts electrons from graphene oxide, reduces the corroded area, and forms cerium ions.” This phrase is not clear (…CeO2-GO is corroded …, cerium oxide (CeO2) accepts electrons …, and forms cerium ions?). Please revise and provide references.
Response: Thank you very much for your professional guidance and valuable advice. We apologize for not expressing the CeO2-GO self-healing mechanism clearly in the previous modifications. This statement has been revised in the manuscript and provides reference materials. Meanwhile, this section has been moved to section 3.5. (Please refer to page 21) .The specific description of the self-healing mechanism of CeO2-GO is as follows:
The CeO2-GO/epoxy nanocomposite coating possesses self-repair capability. This mechanism is based on the interaction between two key materials: cerium oxide and graphene oxide. When the surface of the coating sustains minor damage, the nanoparticles of CeO2 and GO can diffuse and fill the defects and cracks, reacting with the surrounding substrate to form a new protective layer. In this way, CeO2-GO can partially repair itself when subjected to corrosion through the interaction between cerium ions and graphene oxide, thereby maintaining its anti-corrosion properties. This "corrosion self-repair coordination mechanism" provides more reliable and long-lasting protection for anti-corrosion materials, enabling them to be used in more severe environments.
-11- (For the sake of clarity please rewrite the Experimental part. Avoid repetition, and join the information of electrochemical measurements.)
Please write the experimental part in the passive form and separate for quantities unit from the numerical value: 10 g of sodium bicarbonate and 186.6 mg… was dissolved…
Move the paragraph in 2.2: “CeO2-GO has a unique corrosion self-repair coordination mechanism that enables it to partially repair…” into Results and Discussion.
Response: Thank you very much for your professional review. We fully agree with your opinion. In order to express the experimental part more clearly, it has been re expressed in a passive form and the quantity units have been separated from the numerical values. The paragraph in section 2.2: "CeO2 GO has a unique corrosion self-healing coordination mechanism that enables partial repair..." has been moved to section 3.5 of the results and discussion. The specific modifications are as follows:
CeO2-GO nanocomposites were prepared by hydrothermal synthesis method and mixed with epoxy resin to prepare CeO2-GO/EP composite coatings. In 500 ml of deionized water, 10 g of sodium bicarbonate and 186.6 mg of ethylenediaminetetraacetic acid disodium salt dihydrate (EDTA.2Na) were dissolved. The solution was stirred well and placed in a dialysis bag. The solution in the dialysis bag was boiled for 10 minutes. After the dialysis bag was cleaned, the graphene oxide liquid was added, and the mixture was boiled again in a beaker for 10 minutes. Then, the liquid was dissolved in deionized water and dispersed by ultrasonic treatment for 30 minutes. The dispersed solution was transferred into the dialysis bag and stirred in a beaker for 3 days. Finally, the liquid was extracted from the dialysis bag and a graphene oxide dispersion was obtained using an ultrasonic cell disrupter. 0.807 g of cerium nitrate hexahydrate was weighed and dissolved in a beaker containing 4 ml of ammonia solution. The solution was stirred with a glass rod and then transferred into a polytetrafluoroethylene-lined container, which was placed inside a high-pressure reactor. The reactor was placed in an oven set at 160 ℃ for 24 hours. After the reaction was complete, the product was filtered, dried, and ground to obtain CeO2-GO nanoparticles.
-13- L. 167: (Detail on the coating process …).
Inform the type of coating process.
Response: Thank you very much for your professional and constructive comments. We are very sorry that we did not clearly state the type of coating process preparation. Now it has been modified in section 2.2 of the original text. The specific modifications are as follows.
CeO2-GO nanocomposites were prepared by hydrothermal synthesis method and mixed with epoxy resin to prepare CeO2-GO/EP composite coatings. The prepared CeO2-GO composite material were mixed with epoxy resin in a mass ratio of 0.5% and coated on the surface of Q235 steel through a wire rod coating device. (Please refer to page 4)
- 15- (Discuss the results.) Include references in the Results and discussion and in section 3.5.
Response: Thank you very much for your professional review. We fully agree with your viewpoint. References have been added to the results and discussions in the manuscript, as well as in section 3.5.
- 25- (Fig. 9 -16: Present EIS results in a more compact form to make them comparable). There are 6 figures of EIS results. Reduce the number by joining Bode Plots. An EIS reference of bare carbon steel is necessary to evaluate the protection efficiency of the coatings.
Response: Thank you very much for your professional guidance. In order to present the EIS results in a more compact form and make them comparable, the original figures 9-16 have been modified. We strongly agree with your viewpoint that an EIS reference for bare carbon steel is necessary for evaluating the protective efficiency of coatings. Sorry, we did not present EIS results for exposed carbon steel in this study. In our future work, we will consider EIS reference for exposed carbon steel. Your review has constructive guidance for our work. Thank you very much.

Reviewer 2 Report
Dear authors,
You have done a great job in revising the papers. I have no further suggestions.
Best regards.
Author Response
Dear authors,
You have done a great job in revising the papers. I have no further suggestions.
Best regards.
Response:
Dear reviewer:
Thank you very much for taking your time to review this manuscript. We feel great thanks for your professional review work on our article. Your revision suggestions have greatly contributed to the quality of our article. Once again, thank you and wish you all the best.
Kind regards.
